# Vector-borne *Trypanosoma brucei* parasites develop in artificial human skin and persist as skin tissue forms

Christian Reuter[1,2,8], Laura Hauf[1,8], Fabian Imdahl[3,4], Rituparno Sen[3], Ehsan Vafadarnejad[3], Philipp Fey[5], Tamara Finger[5], Nicola G. Jones [1], Heike Walles[5,6], Lars Barquist [3], Antoine-Emmanuel Saliba [3,7], Florian Groeber-Becker[2,5] & Markus Engstler [1] ✉

Transmission of *Trypanosoma brucei* by tsetse flies involves the deposition of the cell cycle-arrested metacyclic life cycle stage into mammalian skin at the site of the fly's bite. We introduce an advanced human skin equivalent and use tsetse flies to naturally infect the skin with trypanosomes. We detail the chronological order of the parasites' development in the skin by single-cell RNA sequencing and find a rapid activation of metacyclic trypanosomes and differentiation to proliferative parasites. Here we show that after the establishment of a proliferative population, the parasites enter a reversible quiescent state characterized by slow replication and a strongly reduced metabolism. We term these quiescent trypanosomes skin tissue forms, a parasite population that may play an important role in maintaining the infection over long time periods and in asymptomatic infected individuals.

Understanding the first steps of an infection is crucial for early intervention to prevent disease establishment and its progression. In a number of vector-borne parasitic diseases, infection is initiated when pre-adapted parasites are injected by the vector into the dermis of the mammalian skin[1–3]. Unicellular parasites of the *Trypanosoma brucei* (*T. brucei*) group cause human and animal African trypanosomiases and are transmitted to the vertebrate host by the tsetse fly[4]. Infection of vertebrates begins when the fly takes a blood-meal, thereby depositing the infectious metacyclic form (MCF) of the parasite into the dermal skin layer. Within the dermis, the cell cycle-arrested MCF are activated, re-enter the cell cycle, and differentiate into proliferative trypanosomes. These parasites morphologically resemble the proliferative mammalian life cycle stage known as the bloodstream form (BSF)[5]. However, the timing and the mechanisms

controlling differentiation to proliferative skin-residing trypanosomes remain elusive.

A subpopulation of tsetse-injected trypanosomes was found to reside and proliferate in the skin at the bite site[6]. The skin is regarded as a reservoir tissue during disease, as trypanosomes have frequently been found in the skin of infected animals and humans, even in aparasitemic (i.e., no trypanosomes detectable in blood) and asymptomatic individuals[7–11]. The skin-dwelling parasites of laboratory mice can be transmitted to tsetse flies[6,7], strongly suggesting a contribution to disease transmission.

Here, we have developed an advanced, highly standardized skin infection model, which recapitulates key anatomical, cellular, and functional aspects of native human skin. Using tsetse flies, we have successfully emulated the natural vector transmission of *T. brucei*

[1]Department of Cell and Developmental Biology, Biocenter, Julius-Maximilians-Universitaet of Wuerzburg, Wuerzburg, Germany. [2]Department of Tissue Engineering and Regenerative Medicine (TERM), University Hospital Wuerzburg, Wuerzburg, Germany. [3]Helmholtz Institute for RNA-based Infection Research (HIRI), Helmholtz Center for Infection Research (HZI), Wuerzburg, Germany. [4]Core Unit Systems Medicine, Julius-Maximilians-Universitaet of Wuerzburg, Wuerzburg, Germany. [5]Translational Center Regenerative Therapies, Fraunhofer ISC, Wuerzburg, Germany. [6]Core Facility Tissue Engineering, Otto-von-Guericke University, Magdeburg, Germany. [7]Institute of Molecular Infection Biology (IMIB), Faculty of Medicine, Julius-Maximilians-Universitaet of Wuerzburg, Wuerzburg, Germany. [8]These authors contributed equally: Christian Reuter, Laura Hauf. ✉e-mail: markus.engstler@uni-wuerzburg.de

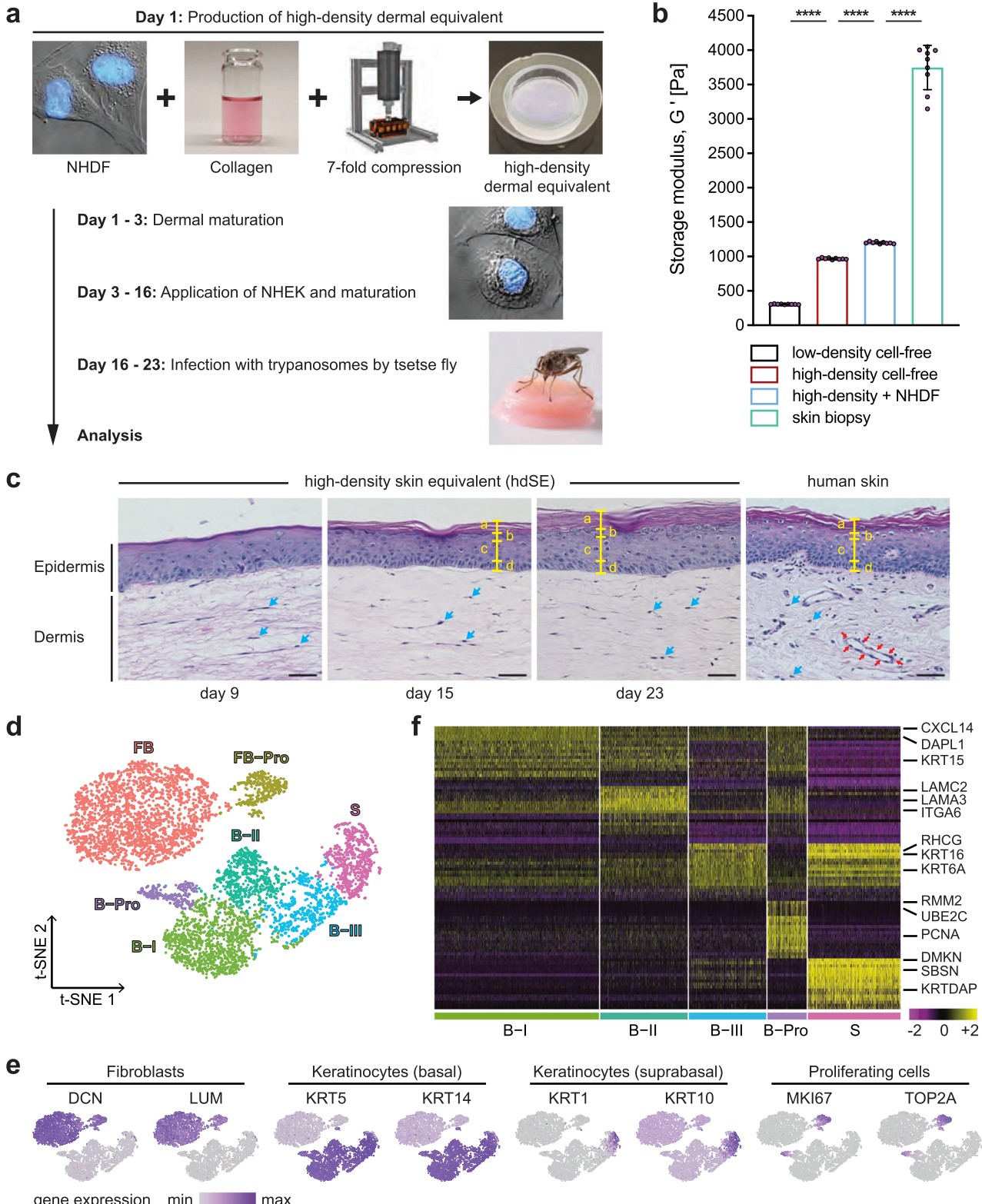

**a** Day 1: Production of high-density dermal equivalent

NHDF + Collagen + 7-fold compression → high-density dermal equivalent

Day 1 - 3: Dermal maturation

Day 3 - 16: Application of NHEK and maturation

Day 16 - 23: Infection with trypanosomes by tsetse fly

Analysis

**b** Storage modulus, G ' [Pa]

□ low-density cell-free
□ high-density cell-free
□ high-density + NHDF
□ skin biopsy

**c** high-density skin equivalent (hdSE) human skin

Epidermis
Dermis

day 9 day 15 day 23

**d** FB, FB-Pro, S, B-II, B-Pro, B-I, B-III

t-SNE 2 / t-SNE 1

**f** CXCL14, DAPL1, KRT15, LAMC2, LAMA3, ITGA6, RHCG, KRT16, KRT6A, RMM2, UBE2C, PCNA, DMKN, SBSN, KRTDAP

B−I, B−II, B−III, B−Pro, S -2 0 +2

**e** Fibroblasts
DCN LUM

Keratinocytes (basal)
KRT5 KRT14

Keratinocytes (suprabasal)
KRT1 KRT10

Proliferating cells
MKI67 TOP2A

gene expression min ▮ max

parasites and observed a rapid activation of the cell cycle-arrested MCF upon arrival in the skin. Unexpectedly, we found that tsetse-transmitted trypanosomes enter, after an initial proliferative phase, a reversible quiescence program in the skin. This skin-residing trypanosome population, here termed skin tissue forms (STF), is characterized by slow replication, a strongly reduced metabolism, and their metabolic activity and transcriptome differ from actively proliferating trypanosomes in the skin and cultivated BSF.

## Results

### Tsetse-transmitted *T. brucei* parasites can infect the skin equivalent

In this study, an advanced primary human skin equivalent (hereafter referred to as a high-density skin equivalent) was developed and validated as a model for vector-borne trypanosome skin infection (Fig. 1a). For this, we developed a computer-assisted compression system (Supplementary Fig. 1a–c) and refined a skin equivalent[12] by improving

**Fig. 1 | High-density skin equivalent has improved mechanical properties and recapitulates key aspects of native human skin. a** Normal human dermal fibroblasts (NHDF) were mixed with reconstituted collagen and compressed sevenfold to generate high-density dermal equivalents (hdDEs). Normal human epidermal keratinocytes (NHEK) were added, and the mature high-density skin equivalents (hdSEs) were infected with trypanosomes by tsetse flies. **b** The storage moduli G' of NHDF-populated and cell-free, high- and low-density dermal equivalents (ldDEs) as well as human skin biopsies indicate improved mechanical properties of hdDEs. The means ± SD of G' were calculated from values in the linear viscoelastic region ($n = 9$, Supplementary Fig. 1d, gray box) determined from at least three individual models/biopsies per condition. Unpaired t-test, two-tailed, **** $p < 0.0001$. Source data are provided as a Source Data file. **c** Hematoxylin and eosin-stained cross sections of hdSEs at different culture times in comparison to native human skin. The yellow markings indicate the individual layers of the epidermis: a, *stratum*

*corneum*; b, *stratum granulosum*; c, *stratum spinosum*; d, *stratum basale*. Blue arrows, NHDF. Red arrows, vascular structure. Scale bar, 40 μm. **d** t-distributed stochastic neighbor embedding (t-SNE) plot of 5958 cell transcriptomes derived from two hdSEs at day 23. The major populations of cell types were fibroblasts (FB), proliferating fibroblasts (FB-Pro), basal (B-I - III), suprabasal (S), and proliferating keratinocytes (B-Pro). **e** The expression of fibroblast and keratinocyte markers from basal, suprabasal, and proliferating cells. Normalized gene expression levels for each cell were color-coded from gray to purple and overlaid onto the t-SNE plot. **f** Heatmap showing the scaled expression levels of the 20 most differentially-expressed genes in each cluster of epidermal keratinocytes. The color key from pink to yellow indicates low to high gene expression levels. Each column represents a single cell, and each row represents an individual gene. Cell-type-specific representative genes are listed to the right.

the mechanical stability of the dermal component to reach a high level of standardization and reproducibility (Fig. 1b, Supplementary Fig. 1d, e, and Supplementary Fig. 2a–c, Supplementary Note). The high-density skin equivalent (hdSE) closely resembled the morphology of 3D human skin (Fig. 1c, Supplementary Note) and expressed known differentiation markers (Supplementary Fig. 2d). In addition, the cellular heterogeneity of the hdSE was analyzed by single-cell RNA sequencing (scRNAseq) (Supplementary Fig. 3a, b). The analysis resulted in a t-SNE plot displaying 7 clusters with distinct expression profiles (Fig. 1d–f, and Supplementary Data 1), which is in good agreement with native human skin[13,14]. Moreover, the analysis revealed the expression of a remarkable repertoire of extracellular matrix-associated genes in the hdSE (Supplementary Fig. 3c).

In order to address whether *T. brucei* could infect the hdSE, different media and primary cells isolated from different human donors were tested to find optimal conditions for co-cultivation of parasites and skin equivalents (Supplementary Fig. 4a–c). To simulate the infection process in the most natural way, tsetse flies were employed to transmit the infective MCF to the hdSEs (Fig. 2a and Supplementary Movie 1). To achieve this, we stacked three hdSEs on top of each other, resulting in a final height of 3 mm. The reason for stacking the hdSEs was due to the proboscis of the tsetse fly, which has a length of 2 mm. In contrast, the hdSEs have a standard height of only 1 mm. Without stacking them, the flies would easily bite through the hdSEs, leading to the majority of the parasites being deposited underneath. The analysis of the skin lesion (Supplementary Fig. 5a) revealed a complex deposition of the parasites within an intricate bite path network in the dermis (Fig. 2b). On average, one fly injected approximately 4000 MCF into the hdSE at different skin depths (Fig. 2c and Supplementary Fig. 5b). Within the observation period of 7 days post-infection (dpi), the tsetse-transmitted MCF increased their total numbers by around two orders of magnitude in the hdSE. Parasite cell death was not apparent (Fig. 2d, e). Scanning electron microscopy revealed a high degree of entanglement of skin-dwelling parasites with collagen fibers (Fig. 2f, I–III), and the trypanosomes were frequently found in close contact with dermal fibroblasts (Fig. 2f, IV–VI). Furthermore, we found that within 1 week, only a minor portion of the parasites (3.4%) was detectable outside the skin equivalent (Supplementary Fig. 8c).

### Metacyclic parasites rapidly differentiate into proliferative trypanosomes in the skin equivalents

At 12 h post-infection (hpi) a 3.6-fold increase of parasites in the S and G2 phases (15.0%) compared to 6 hpi (4.2%) was observed (Fig. 3a and Supplementary Fig. 5c, d). The number of dividing parasites further increased after 18 (19.7%) and 24 hpi (27.1%). Even after 4 and 7 dpi, the skin-residing parasites remained proliferative (25.9% and 24.6%). Within the first 12 hpi the morphology of the injected MCF changed to a BSF-like morphology (Fig. 3b, c). Tracking of single parasites in the skin revealed a significant increase in the mean and maximum swimming speeds within the first 24 hpi (Fig. 3d and Supplementary Movie 2)

comparable to swimming speeds measured in syringe-injected BSF. We next determined the onset of protein synthesis as indication of MCF activation (Fig. 3e, f). While MCF obtained from tsetse flies showed a low protein synthesis rate, a strong 5.4-fold increase was measured in skin trypanosomes within 24 hpi. So far, the results strongly suggested that the tsetse-borne cell cycle-arrested MCF are activated in the hdSE.

### Single-parasite RNAseq reveals the programmed activation of metacyclic forms in the skin equivalents

To define programmed changes in gene expression, the scRNAseq profiles of MCF collected from flies were compared to those of trypanosomes isolated from hdSEs at four different timepoints post-infection (4 hpi, 12 hpi, 24 hpi, and 7 dpi) as well as cultivated BSF. (Fig. 4a and Supplementary Fig. 6a, b). All transcriptomes were examined using an unbiased PCA (Fig. 4b and Supplementary Fig. 6c). Differential gene expression analysis revealed a number of genes specific for each timepoint during parasite differentiation in the skin (Fig. 4c, Supplementary Fig. 6d, e and Supplementary Data 2). Among the differentially-expressed genes, many RNA-binding proteins (RBPs) were up or downregulated at the transcript level for all conditions (Fig. 4d). RBPs play a crucial role in the post-transcriptional regulation of gene expression because transcriptional control of Pol II-transcribed genes does not occur[15,16]. Furthermore, a rapid upregulation of replication-associated transcripts (Fig. 4e) and genes associated with glycolysis (Fig. 4f) was observed. More globally, Gene Ontology (GO) analysis of differentially-expressed genes revealed that distinct sets of GO terms were associated with each timepoint (Fig. 4g, Supplementary Fig. 6f, and Supplementary Data 3). Overall, the scRNAseq results confirmed that MCF are rapidly activated in the hdSEs to give rise to a replicating parasite population.

### Full developmental competence of skin-residing parasites

It has been shown that skin-residing parasites of infected mice can develop into the second mammalian life cycle stage, known as the stumpy form, and that skin-derived parasites can infect tsetse flies[7]. To test if the trypanosomes in the hdSE are competent to differentiate into cell-cycle arrested stumpy forms, infected hdSEs were screened for the presence of parasites expressing the stumpy marker PAD1[17]. At 4 and 7 dpi, PAD1-positive parasites were sporadically detected in the hdSEs (<0.1%; Supplementary Fig. 7a and Supplementary Movie 3). A screening of tsetse flies infected with skin-derived parasites at 1, 4, and 7 dpi revealed that the parasites could infect tsetse flies from 4 dpi, albeit with very low efficiency (Supplementary Fig. 7b).

### *T. brucei* enters a reversible quiescence program in the skin equivalents

In the second phase of skin infection (4 and 7 dpi), the trypanosomes reduced their protein synthesis rate to a basal level (Fig. 3e, f). These parasites continued to proliferate, however slowly (Fig. 2d and Fig. 3a). In addition, replication-associated genes were downregulated at 7 dpi

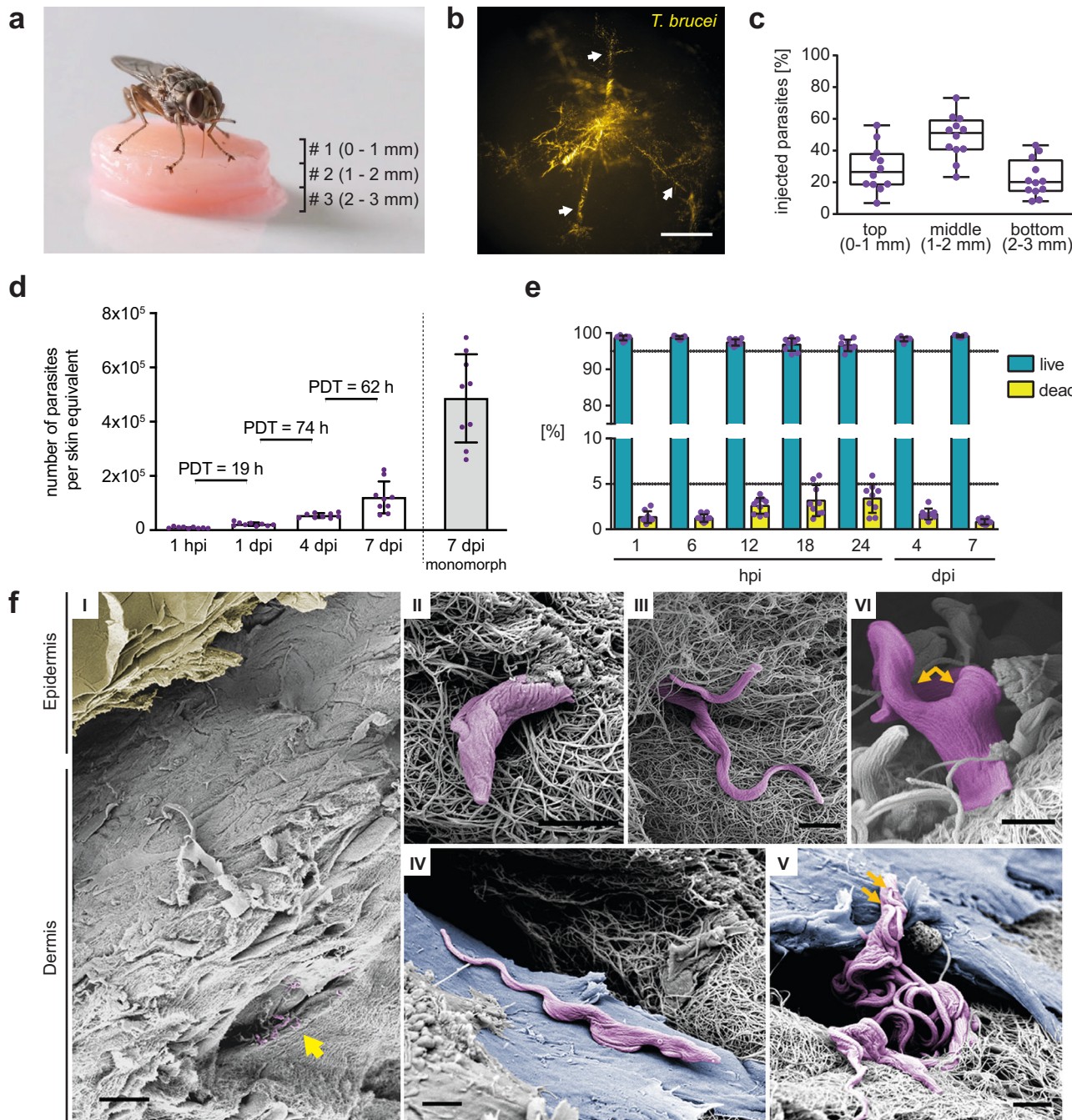

**Fig. 2 | Tsetse-transmitted *T. brucei* parasites establish replicating populations in the skin equivalents. a** Infection of high-density skin equivalents (hdSEs) with tsetse-transmitted trypanosomes. Three hdSEs were stacked (# 1, 2, 3) and exposed to tsetse flies. **b** Trypanosomes were detected in the skin dermis by stereo-fluorescence microscopy. Parasites (yellow) expressing the fluorescent protein tdTomato were found in multiple finger-like lesions (white arrows), probably corresponding to the bite path of the tsetse fly. Scale bar, 500 μm. **c** Analysis of the injection depth of tsetse-transmitted trypanosomes. Three dermal equivalents with a standardized height of 1 mm were stacked and infected with trypanosomes by tsetse flies. The numbers of parasites in each equivalent were quantified after 1 h and expressed as fractions of the total. Results are shown as median ± IQR (*n* = 9 independent infections). Source data are provided as a Source Data file. **d** Cell numbers and population doubling times (PDTs) of trypanosomes in skin equivalents at various times post-infection (hpi, dpi) over a 7-day timecourse. Numbers of monomorphic bloodstream form (BSF) parasites 7 days post syringe injection into

skin equivalents are shown as comparison. Data represent means ± SD (*n* = 9 infected hdSEs examined per time point over three independent experiments). Source data are provided as a Source Data file. **e** Flow cytometry of parasite viability by Calcein-AM staining of infected hdSEs. Results are means ± SD (*n* = 9 independent measurements per time point). The gating strategy is described in Supplementary Fig. 5d. Source data are provided as a Source Data file. **f** Scanning electron microscopy of hdSEs at 4 dpi. (I) Overview showing an intact epidermal layer (yellow) and the presence of trypanosomes (purple) in the connective tissue of the dermis (yellow arrow). Scale bar, 30 μm. (II + III) Entanglement of trypanosomes with collagen fibers. (IV + V) Parasites were found in close contact with dermal fibroblasts (blue). (V + VI) Proliferation was evidenced by the double flagella of trypanosomes (orange arrows). Scale bar, 2 μm (II, III, IV, V, VI). Scanning electron microscopy was conducted on three independently infected hdSEs and representative images are shown.

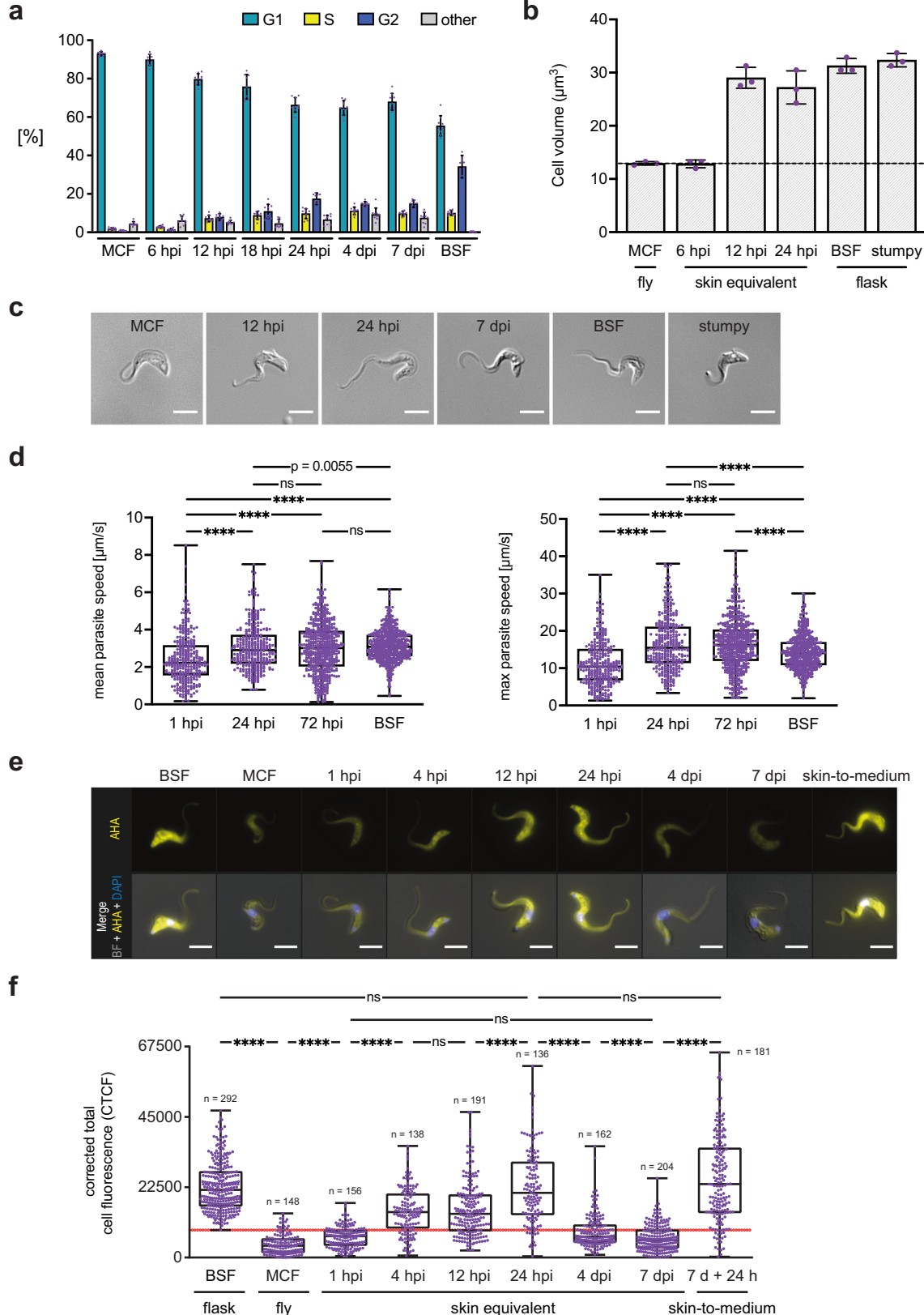

(Fig. 4e) and no significant increase in cell death was observed (Fig. 2e). All these findings are characteristic of quiescent cells[18,19].

As a second independent measurement, the fluorescence reporter tdTomato was used to determine the activity of the rDNA locus as a proxy for metabolic activity[20]. After injection, the tdTomato signal of the skin-residing parasites initially increased within the first 24 hpi, but

then decreased by 2.6- and 4-fold at 4 and 7 dpi, respectively (Fig. 5a and Supplementary Fig. 8a). To test whether this phenotype could be reversed, the migration from the skin to the bloodstream was mimicked by extracting skin-residing parasites on day 7 and transferring them into culture medium. A significantly increased protein synthesis rate (Fig. 3e, f) and tdTomato signal (Fig. 5a and Supplementary Fig. 8a) was detected

**Fig. 3 | Rapid initiation of proliferative growth by tsetse-transmitted trypanosomes post-infection. a** Flow cytometry quantification of cell cycle states using Nuclear Green staining. Metacyclic forms (MCF), skin-resident trypanosomes, and cultured bloodstream forms (BSF) were assayed. Results are means ± SD. 10,000 cells were analyzed per condition in $n = 9$ independent quantification experiments. The gating strategy is described in Supplementary Fig. 5d. Source data are provided as a Source Data file. **b** The cell volume of MCF and skin parasites was compared to BSF and stumpy forms. Results are means ± SD ($n = 3$ independent experiments). Source data are provided as a Source Data file. **c** Morphology of an MCF trypanosome and parasites isolated from high-density skin equivalents (hdSEs) compared to BSF and stumpy forms. Scale bar, 5 μm. Parasites were isolated from hdSEs in nine independent experiments and images of representative cells are shown. **d** Quantification of mean (left) and maximum (right) swimming speeds of individual, skin-resident, naturally transmitted trypanosomes (1 hpi, $n = 319$; 24 hpi, $n = 328$; 72 hpi, $n = 522$), compared to syringe-injected BSF ($n = 532$). Results are shown as median ± IQR. Mann-Whitney U-test, two-tailed, **** $p < 0.0001$, ns, not significant. Source data are provided as a Source Data file. **e** Fluorescence microscopy of trypanosomes isolated from hdSEs in comparison to BSF, MCF, and skin-parasites at 7 dpi transferred to culture medium for 24 h (skin-to-medium). The fluorescence signal of L-azidohomoalanine (AHA) incorporated into nascent proteins is shown in yellow. Cells were counterstained with DAPI (blue) and both channels were merged with the bright field (BF) image. Scale bar, 5 μm. **f** Quantification of protein synthesis in BSF, MCF, skin-resident, and skin-to-medium parasites. The fluorescence intensity of incorporated AHA was measured with ImageJ/Fiji and the corrected total cell fluorescence (CTCF) was calculated. Data are shown as median ± IQR, derived from three independent experiments. Mann-Whitney U test, two-tailed, **** $p < 0.0001$, ns, not significant. The red line defines the threshold for classification as "parasites with low protein synthesis". Source data are provided as a Source Data file.

within 24 h after skin-to-medium transfer. Furthermore, the population doubling time dropped to 7.2 h (Fig. 5b), a value comparable to the 6.2 h of the BSF wildtype strain (Supplementary Fig. 4a). Since protein synthesis is the most energy-consuming cellular process, the percentage of skin-residing parasites with a low protein synthesis rate was quantified for each timepoint. At 4 and 7 dpi, 67% and 76% of skin-residing parasites revealed a strongly reduced protein synthesis rate. Transfer to culture medium reversed this phenotype (Fig. 5c). To exclude that nutrient depletion was causing slow growth in the tissue equivalents, monomorphic BSF were syringe-injected into the skin equivalents. Those cells proliferated rapidly for 7 days, proving that nutrients were not limiting (Fig. 2d).

To test whether the induction of a quiescent state can be attributed to the skin microenvironment, MCF harvested from flies were inoculated directly into culture medium (Supplementary Fig. 8b). Within the observation period of 10 days, the parasites entered exponential growth and revealed a population doubling time of 6.8 h.

To investigate the quiescent trypanosome population in more detail, differential gene expression analysis between parasites isolated from infected skin 24 hpi and 7 dpi was performed (Fig. 5e). The comparative scRNAseq analysis of highly active (24 hpi) and inactive (7 dpi) parasites identified 226 genes that were significantly upregulated at 24 hpi and 99 genes specifically upregulated at 7 dpi (Supplementary Data 4). In addition, the transcriptomes of MCF were compared with parasites isolated at 7 dpi, as both forms possess a reduced metabolism (Fig. 5g and Supplementary Data 4). Among the differentially-expressed genes, the RNA-binding protein ZFP2 (Tb927.11.14950) was downregulated at 7 dpi (Fig. 5d) and could be one factor that regulates the development of a quiescent trypanosome form in the skin.

GO enrichment analysis revealed several metabolic and catabolic processes that were regulated differentially (Fig. 5f, h, and Supplementary Data 3). Specifically, the underlying gene of the GO terms associated with glutamate is a glutamate dehydrogenase (Tb927.9.5900), and it has been shown that glutamate dehydrogenase is induced in quiescent yeast and epithelial cells[21–23].

Lastly, gene expression profiles of quiescent MCF, proliferating parasites at 24 hpi, as well as quiescent trypanosomes at 7 dpi, were compared to flask-cultured BSF (Supplementary Fig. 9 and Supplementary Data 5). GO enrichment analyses were performed (Fig. 5i and Supplementary Data 6) and strongly suggest that neither the 24 hpi parasites, nor the quiescent cells after 7 days are bona fide bloodstream forms. This was further confirmed by investigating the variant surface glycoprotein (VSG) repertoire during colonization of the skin.

Within the first 24 hpi, solely metacyclic VSGs (mVSGs)[24] were found to be expressed in skin-dwelling trypanosomes. At 7 dpi, two isoforms characteristic for BSF, including AnTat 1.1, were identified, while mVSGs continued to be expressed (Fig. 6a, Supplementary Data 7).

Interestingly, AnTat 1.1 was found on the surface of only >0.1% of parasites in the skin models at day 4 and 7 dpi (Fig. 6b), while MCF obtained from flies and cultivated in medium were 17.1% AnTat 1.1-positive after 7 days (Fig. 6c).

In conclusion, we have developed an improved skin tissue model that can be infected with trypanosomes through the bite of the tsetse fly. The parasites almost immediately reactivate the cell cycle and establish a proliferative skin tissue population. In the second phase of skin residence, the trypanosome population shared common features described for quiescent cells. In addition, their metabolism and transcriptome differed from the highly proliferative trypanosomes in the skin and cultivated BSF, and they could be reactivated and returned to an active state. The results demonstrate the existence of a quiescence program without cell cycle arrest in trypanosomes, which may well represent an adaptation to the skin as niche for parasite persistence.

## Discussion

We have established an advanced skin model and validated it through tsetse-borne infection with trypanosomes. We clearly show that this near-natural host environment supports parasite development. Tsetse-transmitted cell cycle-arrested MCF are rapidly activated in the artificial skin and establish a proliferative trypanosome population.

This process is accompanied by (I) reactivation of protein synthesis, (II) re-entry into the cell cycle, (III) acquisition of a BSF-like morphology, and (IV) increased motility.

We found that the parasites re-entered the cell cycle already between 6 and 12 hpi. Moreover, protein synthesis is reactivated very rapidly, after just 1 h, and peaks within 1 day. These findings are consistent with natural infections of mice showing that MCF start multiplying in the skin within 18 h of transmission[6]. Single-parasite RNAseq revealed an upregulation of two RBPs, PUF9, and ZC3H20, within 4 hpi. These proteins have been shown to be involved in replicative and translational processes[25,26]. In addition, we found upregulation of two previously undescribed RBPs, ZC3H35 and ZC3H48, which could well be functional during differentiation of the parasites in the skin. In contrast to the fly-stages, which use proline to feed their tricarboxylic acid cycle, glucose is the only carbon source for ATP production described for replicative BSF[27]. Supporting this, we show that genes associated with glycolysis were rapidly upregulated within 4 hpi in the skin. The number of SMARTseq transcriptomes presented is limited, and there is a possibility that we may have overlooked subtle differences between the cell populations. However, the aim of our experiments was to identify the major differences, which we have achieved.

The morphology of MCF is similar to that of BSF, but their cell body and the free distal part of their flagellum are shorter[28]. After 1 dpi, we observed that the skin-residing parasites exhibited a BSF-like morphology, accompanied by an increased swimming speed. This is consistent with observations that BSF have a higher maximum velocity and flagellar beat frequency compared to MCF[29]. An increased swimming speed could influence the spread of the parasites in the dermis

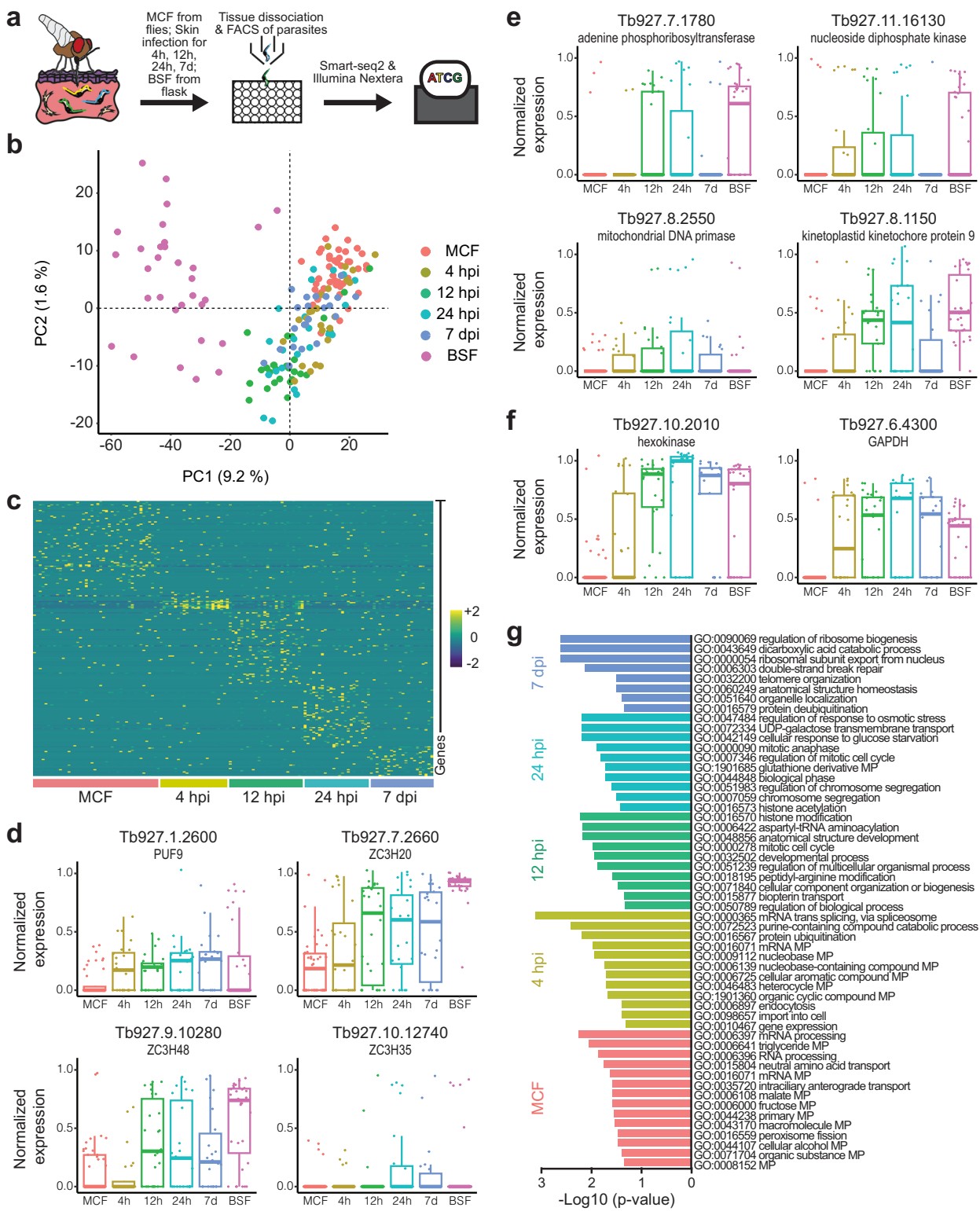

and entry into the draining lymph, where parasites can be detected 1–2 days after infection[30].

The exchange of the VSG coat is a known hallmark of the MCF to BSF transition[31] and the switch from expression of metacyclic to bloodstream form VSGs has been reported to occur between days 4 and 6 post-infection[32,33]. We found expression of metacyclic VSGs in skin-residing parasites even at day 7. VSG AnTat 1.1 is an isoform that is commonly expressed in the AnTat1.1 clone of the pleomorphic

strain EATRO 1125 during the first wave of bloodstream parasitemia, and we indeed found this VSG to be among the expressed VSG mRNAs at 7 dpi. However, positive immunostaining for this VSG isoform was only detected in <0.1% of the skin-residing parasites at 4 and 7 dpi. In contrast, MCF differentiated in suspension culture showed a proportion of 17.1% of cells expressing VSG AnTat 1.1 on the cell surface, suggesting that in culture, probably all parasites have differentiated to BSF.

**Fig. 4 | Single-parasite RNAseq reveals rapid activation of injected metacyclic trypanosomes and early events in skin infection. a** Schematic representation of the experimental pipeline of single-parasite RNAseq. Tsetse-transmitted trypanosomes were isolated from infected high-density skin equivalents (hdSEs) at 4, 12, 24 hpi, and 7 dpi. Single cells of skin-derived parasites, metacyclic forms (MCF) obtained from flies and flask-cultured bloodstream forms (BSF) were sorted into individual microwells using fluorescence-activated cell sorting, followed by processing with the Smart-seq2 and Nextera protocols and subsequent sequencing. **b** Unbiased projection of the first two principal components (PC) of the 170 trypanosome transcriptomes that passed quality control (MCF, $n = 44$; 4 h, $n = 24$; 12 h, $n = 26$; 24 h, $n = 23$; 7 d, $n = 22$; BSF, $n = 31$). Each dot represents a single parasite. Colors indicate the time when the parasites were isolated from hdSEs after infection. **c** Heatmap showing the scaled, time-resolved expression levels of

significantly upregulated genes in MCF and skin-resident parasites as identified by DESeq2 (absolute log2 fold change > 2, adjusted $p$ value < 0.01). Each column represents a single parasite, and each row represents an individual gene. The color key from purple to yellow indicates low to high gene expression levels. Boxplots showing the normalized expression of selected RNA-binding proteins (**d**), replication-associated genes (**e**), and glycolysis-associated genes (**f**) in MCF, parasites isolated from skin, as well as BSF. Results are shown as median ± IQR. The number of cells analyzed for each condition is indicated in (**b**). **g** Biological process-associated Gene Ontology (GO) terms significantly enriched in MCF and parasites isolated from skin. GO terms were filtered with the Revigo webtool to avoid redundancies. Bars represent the level of significance of the term enrichment determined by Fisher's exact test.

---

The VSG coat in proliferating cells can rapidly be exchanged by dilution through cell division. This is not the case for quiescent or slowly dividing cells. Thus, while BSF VSG expression sites might be already activated in some of the STF after day 4, this could just be reflected in the transcriptomes. The few trypanosomes expressing an AnTat 1.1 surface coat are probably not STF, but parasites that had developed to the BSF stage.

Although the reason(s) why trypanosomes persist in mammalian skin remain unknown[6,7], we clearly show that the artificial skin environment strongly influences the parasites. After re-entry into the cell cycle and an initial phase of proliferation, the skin-residing trypanosomes slow down growth, reduce DNA synthesis, and downregulate protein translation and metabolism. Concomitantly, they have very low rates of parasite death. All these characteristics are hallmarks of quiescent cells[18,19,34–36]. We postulate that the initial phase of trypanosome proliferation functions in establishing a robust population in the host skin. Once this is achieved, a second adaptation step yields a slow growing, skin-residing trypanosome population, which contributes to parasite persistence. Differential gene expression analysis revealed that the quiescent parasites clearly differ from the early proliferative trypanosomes after infection and BSF cultured in medium. All the above prompts us to propose that trypanosomes transmitted by the tsetse fly into skin enter a persister-like state, the quiescent skin tissue form (STF). We do not consider the STF as a yet undiscovered life cycle stage of *T. brucei*. We rather suggest that the formation of STF (and probably also the adipose tissue forms (ATF) residing in fat[37,38]) are evolutionary examples for the amazing flexibility in parasite life cycles.

We found that the STF replicate 10-times slower and have a protein synthesis rate 3.5-times lower than cultured BSF. Slow-growing persister cells have been described for a few other protist parasites, such as *Leishmania* or *Trypanosoma cruzi*[19]. In the related protist, *Leishmania mexicana*, quiescent amastigotes in murine lesions show very slow growth with a doubling time of 12 days and low rates of protein turnover[39], and slow-growing intracellular amastigotes of *Leishmania major* with a doubling time of 60 hours have also been reported[40]. In *Trypanosoma cruzi*, chronic infections are associated with a reduced replication rate of the parasite in the colon[41].

Skin-residing trypanosomes in a mouse model can develop into stumpy forms and their proportion has been determined to be 20% on day 11 post-infection[7]. Although we found stumpy forms in our skin model, their proportion was very low with <0.1% of all parasites on days 4 and 7. On the one hand, this shows that the trypanosomes in the skin model are, in principle, capable of stumpy formation at these early timepoints. On the other hand, the STF should be resistant to stumpy induction, as the postulated paracrine induction of stumpy formation generated by trypanosome excreted oligopeptidases[42] would have yielded a much larger proportion of stumpy parasites in the confined skin environment. Alternatively, STF do not secrete oligopeptidases. The hypothesis of specific skin tissue forms is further supported by the very low infection rates of tsetse flies that were fed with skin-derived trypanosomes. We have recently shown that a single bloodstream

stage trypanosome suffices to successfully infect a tsetse fly, and that it does not matter if this is a proliferative BSF or stumpy stage parasite[43]. Thus, if the STF would be BSF, we would expect much higher fly infection rates. This suggests that STF are not tsetse-infective.

In conclusion, our experiments are compatible with the following scenario. The tsetse fly deposits cell cycle-arrested MCF into the host skin. The feeding insect ruptures blood vessels. MCF parasites that directly enter the circulation through the pooled blood, develop to the proliferating BSF stage. Parasites that do not enter the bloodstream but stay in the skin start proliferating in tissue spaces, and in this way establishing a viable population of proliferative trypanosomes within 1 day. At this timepoint they adapt to the skin environment as quiescent STF, which can persist in the host's skin for extended periods. As the skin interstitial fluid is continuous with the lymphatic system, STF parasites will eventually be drained by suction forces into the lymph, where they start proliferating as BSF. The skin acts as a reservoir for African trypanosomes[7,11] and the hidden, quiescent, but motile STF could act as a source of parasites that can continuously repopulate the blood.

Our discovery of STF in a close to nature, tractable tissue model might provide an experimental system for tackling the important question as to how the parasites might be slipping under the radar of the inflammation response. In recent years, there has been increasing evidence that trypanosomes persist in the skin of aparasitemic and asymptomatic individuals[7,11]. It has also been suggested that these individuals contribute to the transmission of the disease[10,44,45]. In addition, case reports of longstanding infections with little to no symptoms[46–49] suggest that at least some individuals may develop latent trypanosome infections, in which quiescent, slowly replicating trypanosomes residing in the skin may play an important role. Furthermore, the skin quiescence might explain some of the treatment failures in humans[50].

Lastly, we would like to emphasize that our work introduces the host-parasite interface as a useful tool for the qualification of human tissue models. The advanced skin tissue model developed in the course of this study is permissive for dual scRNAseq studies of skin cells and parasites during early stages of a natural, vector-borne infection. The in-depth analyses described here revealed the existence of skin tissue forms of *T. brucei* and, at the same time, proved that our skin model delivers all cues required for induction of parasite development. However, it is important to acknowledge that these models do not possess the same level of complexity as native human skin in terms of immune competency, vascularization, and innervation. Furthermore, the in vitro models lack the systemic component. In vivo animal models provide these essential components, albeit with species-specific variations. Therefore, for future studies, it is crucial to carefully consider whether an in vitro or in vivo model is more appropriate for the specific research question at hand. In contrast to laboratory animals, tissue models allow the use of primary human cells and they can be iteratively advanced. The introduction of new cell types, such as macrophages will enable infection studies with other important parasites, for example *Leishmania*.

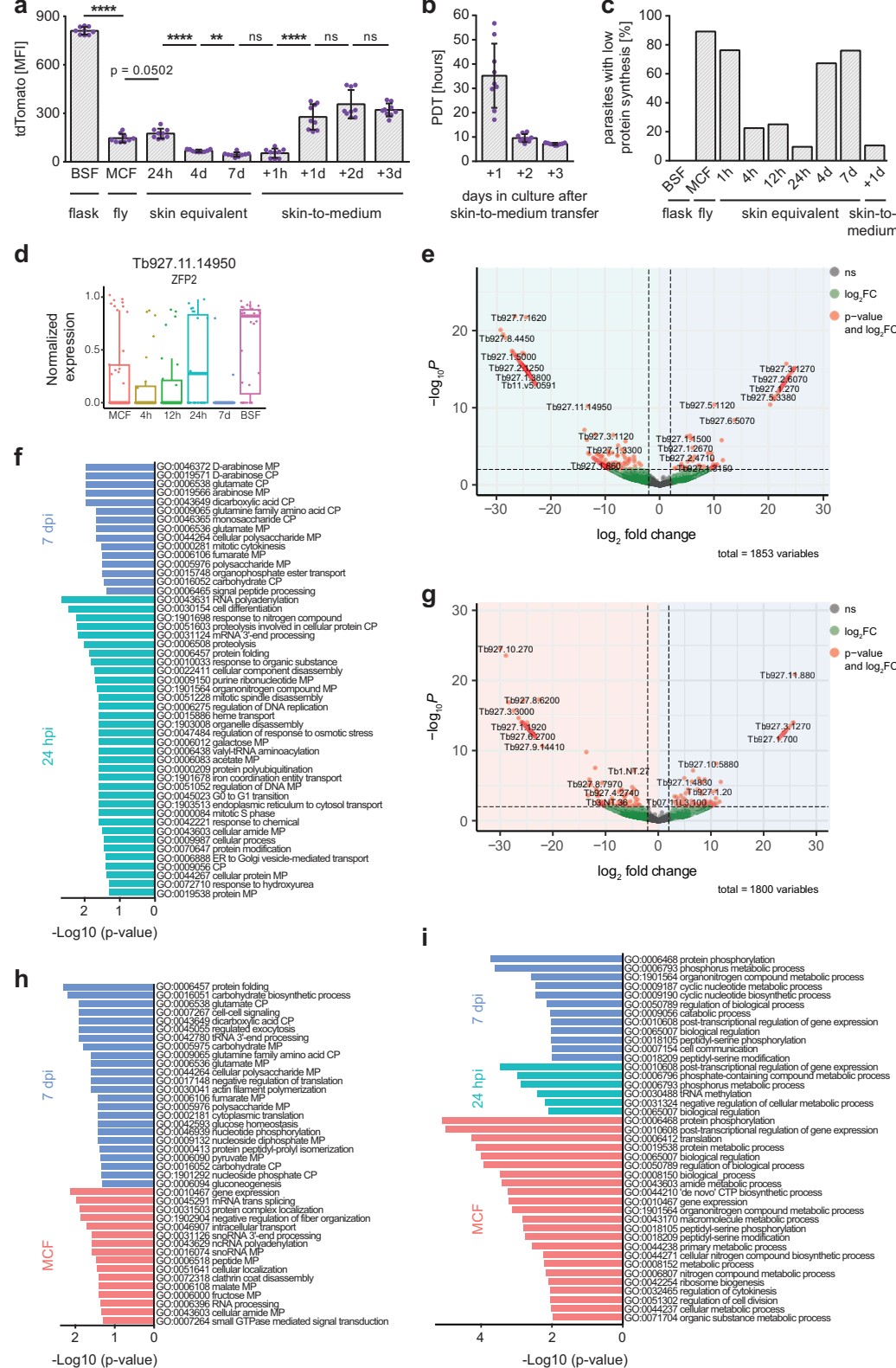

## Methods

### Ethical clearance statement

Normal primary human epidermal keratinocytes (NHEK) and dermal fibroblasts (NHDF) were isolated from biopsies of preputial skin from juvenile donors aged between 2 and 5 years. All donors' legal representative(s) provided full informed consent in writing. The study was approved by the local ethical board of the University of Würzburg (vote 182/10 and 280/18-SC).

### Isolation of primary human cells and culture

Biopsies of preputial skin were washed with PBS containing $Mg^{2+}/Ca^{2+}$ and after removal of excess fat and connective tissue, cut into

**Fig. 5 | Trypanosomes reversibly enter a quiescent state in the skin that is characterized by a unique gene expression profile. a** Flow cytometry of the tdTomato mean fluorescence intensity (MFI) of skin-resident trypanosomes, bloodstream forms (BSF), metacyclic forms (MCF), and 7 dpi skin-parasites transferred to culture medium (skin-to-medium). $1 \times 10^4$ parasites were analyzed per condition. Results are means ± SD ($n = 9$ independent measurements). Unpaired $t$ test, two-tailed, **** $p < 0.0001$, ** $p < 0.01$, ns, not significant. The gating strategy is described in Supplementary Fig. 5d. Source data are provided as a Source Data file. **b** Population doubling times (PDTs) of skin-to-medium parasites. Data represent means ± SD ($n = 9$ independent experiments). Source data are provided as a Source Data file. **c** Percentages of "parasites with low protein synthesis", indicated by the red line in Fig. 3f, in BSF, MCF, skin-resident and skin-to-medium trypanosomes. Data were derived from three independent experiments. Source data are provided as a Source Data file. **d** Boxplot showing the normalized expression of the RNA-binding protein ZFP2 in MCF, parasites isolated from skin and BSF. Results are median ± IQR (MCF, $n = 44$; 4 h, $n = 24$; 12 h, $n = 26$; 24 h, $n = 23$; 7 d, $n = 22$; BSF, $n = 31$). **e, g** Volcano plots illustrating differential gene expression between 24 hpi vs. 7 dpi (**e**) and MCF vs. 7 dpi (**g**). Genes upregulated at 7 dpi are on the right, genes upregulated in either MCF or 24 hpi are on the left. x- and y-axis represent $\log_2$ fold change ($\log_2$ FC) against the adjusted $-\log_{10} p$ value ($-\log_{10}$P); dashed lines indicate the set thresholds. Genes with an absolute $\log_2$ FC > 2 and p-value < 0.01 are considered as differentially expressed. **f, h, i** Biological process-associated Gene Ontology (GO) terms filtered for redundancies with the Revigo webtool. Bars represent the level of significance of the term enrichment determined by Fisher´s exact test. GO terms enriched in parasites isolated from skin at 7 dpi vs. 24 hpi (**f**) or vs. MCF (**h**). (**i**) GO terms enriched in MCF or parasites isolated from skin at 24 hpi and 7 dpi, each in pairwise comparison with BSF. GO terms enriched in BSF are provided as Supplementary Data 6.

3 mm × 10 mm sized pieces. Subsequently, pieces were incubated with 2 U/ml dispase at 4 °C overnight and the epidermis was separated from the dermis with forceps. To isolate NHEK, epidermal pieces were minced and incubated with 0.05% trypsin/EDTA at 37 °C for 5 min, followed by thorough resuspension to release cells. A 100 μm cell strainer was used to filter cell suspension and keratinocytes were cultured in EpiLife basal medium supplemented with 1% human keratinocyte growth supplement (HKGS) and 1% penicillin/streptomycin. To isolate NHDF, dermal pieces were minced and incubated with 500 U/ml collagenase type XI at 37 °C for 45 min. Dermal pieces were washed with DMEM supplemented with 10% fetal calf serum (FCS), 1% non-essential amino acids (NEAA), and 1% penicillin/streptomycin and transferred into cell culture flasks. Dermal pieces were removed after 5 days after fibroblasts had grown out.

For both cell types, medium was changed every 2–3 days and cells were passaged when reaching 70–90% confluency by Accutase- (NHEK) or Trypsin-treatment (NHDF). To generate dermal equivalents or high-density skin equivalents cells from passages 3 to 5 were used.

## High-density primary human skin equivalent

The first step in the generation of high-density primary human skin equivalents (hdSEs) involves the production of the high-density dermal equivalent (hdDE). The compression reactor was assembled and the membrane of 12 Snapwell™ inserts was perforated with 42 microneedles to improve permeability before the inserts were placed in the designated slots in the reactor. Next, 800 μl of reconstituted collagen solution (collagen = 6.7 mg/ml) containing 45,200 NHDF was filled in each of the 12 compression chambers. Briefly, 8 ml of cold 10 mg/ml collagen type I from rat tail dissolved in 0.1% acetic acid was taken up in a 20 ml syringe without air bubbles using a sterile 13G cannula. Subsequently, $6.78 \times 10^5$ NHDF were resuspended in 4 ml of cold neutralization solution (2x DMEM, 3% FCS, 3% 3 M HEPES, 1% chondroitin sulfate, pH = 8.5) and the suspension was taken up in a 20 ml syringe using a sterile 18 G cannula. Next, both syringes were connected to a three-way valve and mixed by alternately pressing the two syringes. After 8–10 mixing steps, the empty syringe was disconnected and replaced by a Safeflow® needlefree injection port. A 10 ml Combitip® connected to an Eppendorf Multipette M4 was inserted into the injection port and by pressing the syringe the neutralized NHDF-containing collagen solution was transferred into the Combitip®. 800 μl were dispensed air bubble-free in each compression chamber and to ensure complete gelation of collagen, the reactor was incubated 30 min under a sterile workbench and an additional 30 min in a cell culture incubator at 37 °C and 5% CO₂. The lid of the reactor was attached to the linear motor and the solidified collagen gels were compressed by a factor of 7 by moving the lid downwards at constant $2 \times 10^{-6}$ m/s until the reactor was completely closed. The reactor was opened and compressed air was introduced via the adapter into the lid and the compression punches to release the compressed dermal equivalents without damage. Each hdDE had a calculated collagen concentration of 46.9 mg/ml and was characterized by a diameter of 12 mm and a height of 1 mm, resulting in a volume of 113 μl. hdDEs were cultured with DMEM supplemented with 10% FCS, 1% NEAA, and 1% penicillin/streptomycin. On day 3, 250 μl of EpiLife basal medium supplemented with 1% HKGS, 1% penicillin/streptomycin, and 1.44 mM CaCl₂ (= E2 medium) containing $5 \times 10^5$ NHEK were added to the apical side of each hdDE to generate the epidermis of the hdSE. The next day, the E2 medium was aspirated and air-liquid interface (ALI) culture was initiated with E2 medium supplemented with 10 ng/ml human keratinocyte growth factor and 252 μM L-Ascorbic acid 2-phosphate (= E3 medium). ALI culture was continued until day 23 at 37 °C and 5% CO₂ with medium change every 2–3 days.

## Trypanosome strains and culture

The monomorphic *Trypanosoma brucei* Lister 427 MITat 1.2 13–90 strain[51] transfected to express a fluorescent tdTomato reporter, was used for skin equivalent infection through syringe-injection. The tdTomato gene is targeted to the ribosomal DNA locus and is constitutively expressed[52]. Monomorphic bloodstream forms were cultured in suspension to a maximum cell density of $1 \times 10^6$ cells/ml in HMI9 medium supplemented with 10% FCS.

The pleomorphic *T. brucei* EATRO 1125 AnTat 1.1 strain harboring the two reporters *GFP::PAD1UTR* and *tdTomato* was used for all other experiments. The construct *GFP::PAD1UTR* is a reporter for the stumpy form of the trypanosomes, in which the green fluorescent protein GFP is coupled to the 3'UTR of the stumpy marker PAD1[17]. The GFP protein is targeted into the nucleus via a nuclear localization signal. The fluorescent protein tdTomato is located in the ribosomal DNA locus and expressed in all life cycle stages of the parasite[52]. Pleomorphic bloodstream forms were cultured in suspension below $5 \times 10^5$ cells/ml in viscous HMI9 medium supplemented with 10% FCS and 1.1% methylcellulose[53]. All cells were cultured at 37 °C and 5% CO₂.

## Tsetse fly colony

Tsetse flies of the subspecies *Glossina morsitans morsitans* were kept in an insectary at 27 °C and a relative humidity of 70% and fed three times a week through a silicon membrane with preheated defibrinated sterile sheep blood.

## Compression reactor design and fabrication

The computer-aided design of the compression reactor was conducted with SolidWorks® and the reactor was manufactured from polyether ether ketone. The reactor allowed simultaneous compression of 12 collagen gels in Snapwell™ inserts. A perforated plate beneath the insert holder prevented the insert membranes from rupturing during compression and a liquid collecting tray collected the displaced water from the gels. The compression chamber increased the compression volume per insert up to 2.8 ml, and all parts could be firmly connected to the compression chamber by tightening the swing screws located on both sides of the liquid collecting tray. The 12

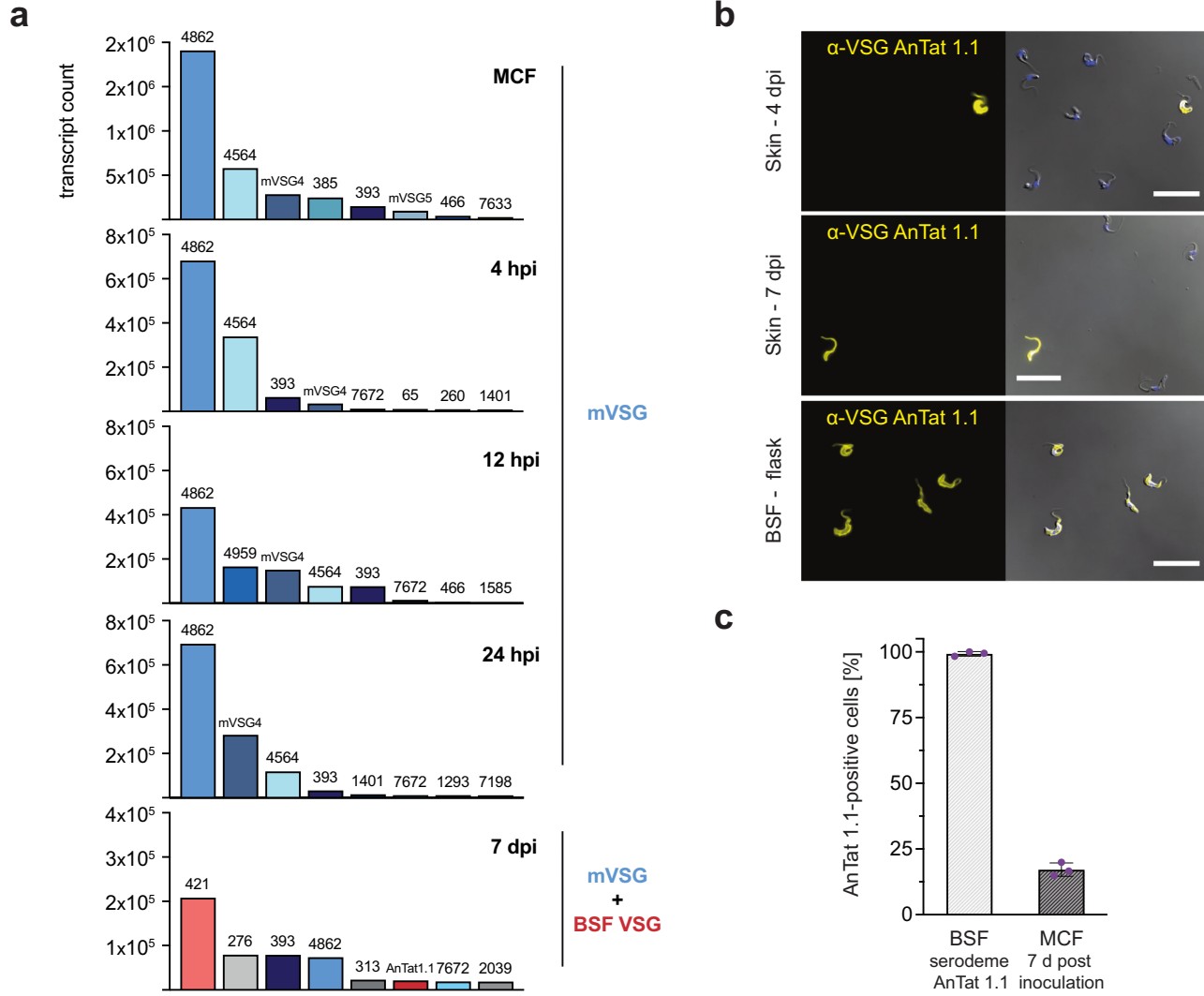

**Fig. 6 | STF continue to express mVSGs alongside BSF VSGs until at least 7 days post infection. a** Expression of the eight most abundant variant surface glyco-protein (VSG) mRNAs found in metacyclic forms (MCF) and skin-resident parasites 4, 12, and 24 hpi and 7 dpi. Within the first 24 hpi, exclusively metacyclic VSGs (mVSGs) are expressed. At 7 dpi, mVSGs are expressed together with bloodstream form (BSF) VSGs. Bar colors indicate confirmed mVSGs, (blue), BSF VSGs (red) and unassigned VSGs (gray). **b** Immunofluorescence microscopy of skin-derived try-panosomes at 4 and 7 dpi, stained using an antibody against the BSF-specific VSG AnTat 1.1 (yellow) and compared to BSF cultured in flasks. On the right, the blue

(DAPI) and yellow (VSG) channels are shown as overlays with the DIC image. Scale bar, 20 μm. Immunofluorescence staining was performed independently three times and representative images are shown. **c** Quantification of VSG AnTat 1.1 expression in BSF of the serodeme AnTat 1.1 and MCF grown for 7 days in HMI9 medium. MCF were obtained from tsetse flies infected with stumpy forms of the serodeme AnTat 1.1. Shown is the percentage of parasites that stained positive for AnTat 1.1. The graph represents means ± SD (*n* = 3 independent quantifications). Source data are provided as a Source Data file.

cylindrical compression punches of the lid were designed to fit exactly into the 12 openings of the compression chamber with a minimum of friction. Due to their defined length the distance between the insert membranes and the lower edge of the compression punches was always exactly 1 mm after complete compression. An adapter for the attachment to the linear motor with a compressed air connection could be mounted centrally on the lid via a screw thread. Via the connection compressed air with a pressure of 4–6 bar could be introduced and was distributed through the sealed cavity in the lid into the 0.5 mm diameter channels, which pass centrally through each of the 12 compression punches. The compressed air was needed to release the compressed collagen gels in the Snapwell™ inserts from the compression punches without damage. Compression was performed with a linear motor from the company NTI AG LinMot & MagSpring (Switzerland). The linear motor was attached to a stainless-steel motor mount via a motor flange and connected to a computer via a servo drive. The motor was controlled using the LinMot®-Talk software.

### Skin dissociation and quantification of cells

To isolate NHDF and trypanosomes from skin equivalents, the epidermis was first removed from the dermis with forceps. The dermis was cut into small pieces with a scalpel and incubated in 1 ml of trypanosome dilution buffer (TDB; 5 mM KCl, 80 mM NaCl, 1 mM MgSO₄, 20 mM Na₂HPO₄, 2 mM NaH₂PO₄, 20 mM glucose, pH = 7.6) containing 250 μg Liberase TL in a water bath at 37 °C for 45 min. Cells were collected by centrifugation at 2000 *g* for 2:30 min, resuspended in TDB, and counted.

### Flow cytometry

Skin equivalents were dissociated, the cells were resuspended in 500 μl TDB and then filtered through a 35 μm cell strainer. 20 min and 5 min before analysis, 2 μM Calcein AM Violet and 2 μM NuclearGreen were added, respectively. Samples were then processed on a FACSAria III cytometer (BD Biosciences). 10,000 trypanosomes were analyzed by gating first on the tdTomato signal to exclude NHDF and extracellular

matrix proteins from analysis. Next, viable parasites were selected by setting a gate on the Calcein AM signal and finally, cell cycle distribution was determined by NuclearGreen fluorescence.

## Determination of trypanosome cell volume

Infected skin equivalents were dissociated, and cells were resuspended in TDB. BSF and stumpy form cells were harvested from culture flasks and MCF were obtained from tsetse flies through salivation into medium. Cells were centrifuged at 1400 $g$ for 15 min at 37 °C. The pellet was resuspended in TDB, cells were counted and added to isotonic solution in a final concentration of $5 \times 10^4$ cells/ml. The isotonic solution was filtered through a 35 μm cell strainer twice before being analyzed with a Multisizer 4e particle sizer and counter (Beckman Coulter). A volume of 500 μl per sample was measured via the coulter principal (electrical zone sensing) using a 50 μm measuring cell. Background from fly saliva, methylcellulose and skin equivalents was mathematically subtracted from the data and peak cell volumes were determined using Prism version 7 (GraphPad).

## Manual injection of trypanosomes into skin equivalents

BSF cultures were collected by centrifugation at 1400 $g$ for 15 min at 37 °C. Cells were resuspended in HMI9 and cell numbers were determined. Cells were pelleted in the picofuge for 2 min and resuspended at a concentration of $5 \times 10^6$ cells/ml. 5 μl of cell suspension (= $2.5 \times 10^4$ cells) were injected into each skin equivalent using a NanoFil™ microinjection syringe connected to a 35 G needle (World Precision Instruments). Infected skin equivalents were cultured on sterile Whatmann® paper at the air-liquid interface with 7 ml of INF medium at 37 °C and 5% CO$_2$. Medium was changed every 2–3 days.

## Infection of tsetse flies

Tsetse flies were routinely infected with stumpy stage trypanosomes within 72 h post-eclosion. BSF were grown to a density of $5 \times 10^5$ cells/ml and cultured for another 48 h without dilution to induce stumpy formation by density. Stumpy forms were harvested by diluting cultures 1:4 with TDB and subsequent filtration to remove methylcellulose, followed by centrifugation at 1400 $g$ for 15 min at 37 °C. Parasites were resuspended in sheep blood at a concentration of $4–8 \times 10^6$ parasites/ml, supplemented with 60 mM N-acetyl-D-glucosamine and 12.5 mM glutathione, and fed to flies. Positive flies were selected by a salivation test (mature salivary gland infection) after 5 weeks.

To infect flies with skin-derived trypanosomes, infected skin equivalents were dissociated and viable parasites were sorted based on their tdTomato and Calcein AM signal with a FACSAria III cytometer (BD Biosciences) to exclude NHDF and extracellular matrix proteins. Parasites were resuspended in sheep blood at $5 \times 10^4$ cells/ml and fed to teneral tsetse flies with their first blood meal. Flies were dissected after 5 weeks and screened for trypanosomes present in the midgut, proventriculus, and salivary glands.

## Natural infection of skin and culture

On day 15 of skin equivalent culture, the E3 medium was exchanged with infection medium (INF; 1:1 mixture of E3 and HMI9 medium supplemented with 1% Anti/Anti, 1%penicillin/streptomycin, and 440 μM CaCl$_2$). Skin equivalents aged between 16 and 23 days, depending on further experiments and duration of culture, were removed from the Snapwell™ inserts. Three skin equivalents were stacked on top of each other on an aseptic microscope slide on a heating plate set to 37 °C and six tsetse flies with a mature salivary gland infection (SG+) were allowed to bite into the stack. In order to have sufficient quantities and equal numbers of parasites for downstream analysis per skin equivalent, the positions were changed and another round of 6 SG+ flies were allowed to bite into the stack. This was repeated three times, so that each of the three skin equivalents was once on each position (bottom, middle, top). Stacking was necessary

because the proboscis of the tsetse fly has a length of 2 mm, whereas the hdSEs have a standard height of 1 mm. Without stacking, the flies simply bit through the hdSEs and deposited most of the parasites underneath. Infected skin equivalents were cultured on sterile Whatmann® paper at the air-liquid interface with 7 ml of INF medium at 37 °C and 5% CO$_2$. The medium was changed every 2–3 days.

## Assessment of injection depth

Cell-free high-density dermal equivalents were used since no further culture was carried out. Three equivalents were stacked on top of each other and 18 SG+ tsetse flies were allowed to bite into the stack without rotation. Immediately after infection, dermal equivalents were dissociated and trypanosomes were counted by flow cytometry based on their tdTomato signal. Since each high-density dermal equivalent had a standardized height of 1 mm, a depth of up to 3 mm could be simulated with a resolution of 1 mm.

## Assessment of protein synthesis rate

Skin equivalents were dissociated and trypanosomes were resuspended in 500 μl of prewarmed methionine-free RPMI containing 10% FCS, 1% Anti/Anti, 1% penicillin/streptomycin, and either 50 μM L-Azidohomoalanine (AHA) or 50 μM L-methionine as negative control. As a further control, BSF from cell culture flasks were included in each experiment. All samples were incubated in a cell culture incubator for 1 h at 37 °C. Subsequently, cells were centrifuged at 2000 $g$ for 2:30 min, resuspended in 500 μl TDB supplemented with 2 μM Calcein AM Violet and filtered through a 35 μm cell strainer. Viable parasites were sorted based on their tdTomato and Calcein AM signal with a FACSAria III cytometer (BD Biosciences) to exclude NHDF and extracellular matrix proteins. $1–5 \times 10^4$ parasites were sorted in 24-well plates containing coverslips pre-coated with poly-L- lysine (Ø 15 mm) and filled with 500 μl 3% PFA (125 μl TDB + 375 μl 4% PFA). 24-well plates were centrifuged at 1500 g for 5 min to attach parasites to the coverslips and washed once with 500 μl PBS containing 3% BSA (PBS-B). Trypanosomes were permeabilized with 500 μl PBS-B containing 0.5% Triton X-100 for 20 min and washed twice with 500 μl PBS-B. 300 μl of Click- iT reaction cocktail (prepared according to the manufacturers' instructions) were added to each cover slip and incubated for 30 min at room temperature (RT) protected from light. Following a further washing step with 500 μl PBS-B, the coverslips were carefully taken out of the cavities of the 24-well plate and mounted upside down on a microscope slide with 7.5 μl Fluoromount-G with DAPI. At least 136 trypanosomes from three individual experiments were imaged per timepoint with a DMI6000B wide-field fluorescence microscope (Leica), equipped with a 100× oil objective (NA 1.4) and a DFC365FX camera (pixel size 6.45 μm). Images were taken by focusing in the DAPI channel on the cell nucleus and kinetoplast. Subsequently, images were taken in brightfield, DAPI and AHA channels. Images were then analyzed with the ImageJ/Fiji software. To quantify the fluorescence intensity of incorporated AHA or L-methionine, a rectangular region of interest (ROI) was drawn around individual parasites. The area, mean fluorescent intensity, and integrated density of the ROI were measured in the AHA channel along with several adjacent background measurements using the built-in measurement program (Analyze > Measure). The corrected total cell fluorescence (CTCF) of individual cells incubated with either AHA or L-methionine was calculated using the following formula: CTCF = integrated density − (area of selected cell × mean fluorescence of background). In addition, to correct for cell autofluorescence, the CTCF values of parasites incubated with L-methionine (negative control) were subtracted from all CTCF values of parasites incubated with AHA.

## scRNAseq of trypanosomes

After natural infection, skin equivalents were cultured for another 4 h, 12 h, 24 h, and 7 d, respectively. In addition, freshly collected MCF

obtained from tsetse flies harvested by continuous salivation into medium and flask-cultured BSF were used. To ensure consistency, the experimental procedures involved in sample preparation for RNA sequencing (centrifugation, cell sorting by FACS, and cell lysis) were performed on the MCF and BSF following the same protocols used for the trypanosomes isolated from the skin. Skin equivalents were dissociated and single viable parasites were sorted based on their tdTomato and Calcein AM signal with a FACSAria III cytometer (BD Biosciences) into 48-well plates containing 2.6 µl of 1× lysis buffer (Takara) supplemented with 0.01 µl of RNase inhibitor (40 U/µl; Takara). Immediately after sorting, cells were placed on ice for 5 min and stored at −80 °C. Library preparation and sequencing were carried out as described previously[54]. Briefly, 0.2 µl of a 1:20 × 10⁶ dilution of ERCC Spike-in Control Mix 1 (Thermo Fisher Scientific) were added to the lysates of single parasites and libraries were prepared using SMART-Seq v.4 Ultra Low Input RNA Kit (Takara) using a quarter of the reagent volumes recommended by the manufacturer. 27 cycles were used for PCR amplification and cDNA was purified using Agencourt AMPure XP beads (Beckman Coulter) using 15 µl of elution buffer (Takara). Library quantification was performed with a Qubit 3 Fluorometer with dsDNA Hs Assay kit (Life Technologies) and the quality of the libraries was assessed using a 2100 Bioanalyzer with High Sensitivity DNA kit (Agilent). 0.5 ng of cDNA was subjected to a tagmentation-based protocol (Nextera XT, Illumina) using a quarter of the recommended volumes, 10 min for tagmentation at 55 °C and 1 min extension time during PCR amplification. Libraries were pooled and sequencing was performed in paired-end mode for 2 × 75 cycles using Illumina's NextSeq 500.

## Analysis of scRNAseq data of trypanosomes

After demultiplexing, data quality was examined using FastQC (version 0.11.7). Illumina adaptors were removed using cutadapt (version 3.2). Trimmed reads were mapped to the TREU927 (version 48) genome with the ERCC spike-in sequences included, using RNASTAR (version 2.6.1b). Read counts for each gene were determined using featureCounts (version 2.0.1) and genes with ≥5 aligned reads were considered detected. For MCF, 4 h, 12 h, 24 h, and 7 d, only those scRNAseq datasets with >200 and <1000 detected genes and a library size between $2 \times 10^5$ and $2 \times 10^6$ reads were considered for analysis. For BSF, the upper threshold of <1000 genes per scRNAseq dataset was raised to <2100 due to the overall higher number of detected genes. This difference between the BSF data and the other data sets was considered in all downstream analyses. Principal component analysis (PCA) was performed selecting the 1000 genes with the highest variance. The following libraries were excluded from the analysis due to BARP gene expression: "0h_1_B5_1", "0h_1_G2", "0h_1_C5_1". In total, 14 genes were considered as BARP genes: "Tb927.9.15510", "Tb927.9.15520", "Tb927.9.15530", "Tb927.9.15540", "Tb927.9.15550", "Tb927.9.15560", "Tb927.9.15570", "Tb927.9.15580", "Tb927.9.15590", "Tb927.9.15600", "Tb927.9.15610", "Tb927.9.15620", "Tb927.9.15630", "Tb927.9.15640".

Differential gene expression analysis was conducted with DESeq2 (version 1.30.0) and SCDE (version 2.18.0; only used in early vs. late analysis). Features with an absolute log2 fold change > 2 and adjusted P value < 0.01 (DESeq2) or a z-score >1.96 (SCDE) were considered as differentially expressed, respectively.

For analyzing the VSG expression in MCF, 4 h, 12 h, 24 h, and 7 d data sets, reads were mapped to the VSGnome of AnTat 1.1E (EATRO1125) including flanks and counted using kallisto | bustools[55]. Since there was an extensive overlap between the sequences reported in the VSGnome, these were collapsed first using CD-HIT[44,56] to remove sequence redundancy. Out of the VSGs found to be expressed, the top 20 for each time point are listed in Supplementary Data 7.

## Gene Ontology (GO) analysis

GO term enrichment within the differentially expressed genes between the six data sets (MCF, 4 h, 12 h, 24 h, 7 d, and BSF) were determined by Fisher's exact test using the GO enrichment tool on the TriTrypDB webserver (https://tritrypdb.org/tritrypdb/app)[57] and summarized using the REVIGO webtool[58] to avoid redundancies. For early vs. late GO term analysis, genes with a z-score >1.28 (SCDE) were used for enrichment.

## Tracking of single parasites

Individual trypanosomes were monitored in the skin equivalents based on the signal of the fluorescence reporter tdTomato using a fluorescence stereomicroscope (MZ16 FA, Leica). Movies were acquired for 5 min with a 5× dry objective and a rate of 4 frames per second. For tracking of single parasites movies were analyzed with Imaris software. Only objects with a diameter larger than 6 µm and slower than 80 µm/s were considered as individual trypanosomes. The maximum gap size was set to 20 frames and tracks shorter than 60 s were excluded from analysis. All tracks were validated manually and, if necessary, incorrect tracks or incorrectly assigned objects were corrected.

## Immunofluorescence staining of trypanosomes

Skin equivalents were dissociated, and $5 \times 10^4$ trypanosomes were sorted on coverslips pre-coated with poly-L-lysine and fixed with 3% PFA as described above. Parasites were blocked with 500 µl PBS-B for 30 min and incubated with a rat anti-VSG AnTat 1.1 antibody (Davids Biotechnologie) diluted 1:4000 in PBS-B for 1 h at RT, followed by two wash steps with PBS-B and incubation with an anti-rat IgG Alexa 647 secondary antibody (A21472; Thermo Fisher Scientific) diluted 1:500 in PBS-B for 30 min at RT. Cells were washed twice with PBS-B and coverslips were mounted upside down on a microscope slide with 7.5 µl Fluoromount-G with DAPI. Images were acquired with a DMI6000B wide-field fluorescence microscope (Leica).

## Scanning electron microscopy

Dermal equivalents were fixed in Karnovsky solution (2% PFA, 2.5% glutaraldehyde in 0.1 M cacodylate buffer, pH = 7.4) overnight at 4 °C. Specimens were washed three times for 5 min at 4 °C with 0.1 M cacodylate buffer, followed by incubation for 1 h at 4 °C in post-fixation solution (2.5% glutaraldehyde in 0.1 M cacodylate buffer, pH = 7.4). Samples were washed, incubated in cacodylate buffer containing 2% tannic acid and 4.2% sucrose for 1 h at 4 °C, and washed three times with water for 5 min at 4 °C. Subsequently, specimens were dehydrated in an ascending acetone sequence, critical point dried and coated with gold-palladium. Images were acquired with a JEOL JSM-7500F scanning electron microscope using the detector LEI for secondary electrons at 5 kV.

## Dermal contraction, weight, and viability

To determine dermal contraction and weight loss, the dermal area and weight were measured regularly during culture. Therefore, individual low- and high-density dermal equivalents were cultured in petri dishes (35 mm) containing 3 ml of DMEM supplemented with 10% FCS, 1% NEAA, and 1% penicillin/streptomycin. For up to 3 weeks, each dermal component was photographed once a week using graph paper and the weight was determined with an analytical balance after the medium was completely aspirated. After the measurements, fresh medium was added and the dermal equivalents were cultured further.

Fibroblast viability after compression was determined by fluorescein diacetate (FDA) and propidium iodide (PI) staining. Therefore, dermal equivalents were submersed with 3 ml PBS containing 20 µM FDA and 3 µM PI for 5 min at 37 °C and subsequently washed three times with PBS-. Images were acquired using a MZ16 FA fluorescence stereomicroscope (Leica).

## scRNAseq of skin equivalents

The dermis of two individual hdSEs at day 23 was separated from the epidermis with forceps and subsequently cut into small pieces and

incubated in 1 ml of TDB containing 250 µg Liberase TL for 45 min at 37 °C. To isolate NHEK the epidermis was incubated in 3 ml of PBS- with gentle agitation. PBS- was aspirated, the epidermis cut into small pieces and 1 ml of 0.05% Trypsin/EDTA was added for 15 min at 37 °C. 100 µl of FCS were added and NHEK was released by thorough resuspension. Cells were collected by centrifugation at 300 g for 5 min, resuspended in 500 µl INF medium containing 0.5 µg DAPI and filtered through a 35 µm cell strainer. Viable NHDF and NHEK were sorted based on their negative DAPI signal with a FACSAria III cytometer (BD Biosciences) and collected in a single tube. Cell density was evaluated by microscopy using a hemocytometer and adjusted to 1000 cells/µl.

Subsequently, the Chromium Next GEM Single Cell 3′ Reagent Kits v3.1 (10x Genomics) were used for reverse transcription, cDNA amplification, and library construction according to the manufacturers' instructions. 16.5 µl of cell suspension were used to recover 10,000 cells and libraries were quantified with a Qubit 3 Fluorometer (Thermo Fisher Scientific). Quality was checked using a 2100 Bioanalyzer with High Sensitivity DNA kit (Agilent). Sequencing was performed in paired-end mode with a SP flow cell (2 x 50 cycles) using NovaSeq 6000 sequencer (Illumina).

### Analysis of scRNAseq data of skin equivalents

CellRanger v3.1.0 (10x Genomics) was used to process scRNAseq data. To generate a digital gene expression matrix, reads were mapped to the human reference genome GRCh38 and the number of UMIs for each gene in each cell was recorded. Seurat R package (version 3.1.4)[59] was used for further analysis of scRNAseq data. Only cells with >2500 genes, between $10^4$ and $10^5$ UMIs, and between 1% and 15% mitochondrial (mt) genes were considered for downstream analyses. The "NormalizeData" function was used to normalize sequencing reads of each gene to total UMIs in each cell. The "ScaleData" function was used to scale and center expression levels in the data set for dimensional reduction. The top 2000 features that exhibit high cell-to-cell variation in the dataset were selected with the function "FindVariableFeatures" and used for dimensionality reduction. Total cell clustering was performed by "FindNeighbors" function using the first 20 dimensions and "FindClusters" function at a resolution of 0.4. Dimensionality reduction was performed with "RunTSNE" function and visualized by t-Distributed Stochastic Neighbor Embedding (t-SNE). Marker genes for each cluster were determined with the Wilcoxon rank-sum test by "FindAllMarkers" function. Only those with |"avg_logFC"| > 0.25 and "p_val_adj" <0.05 were considered as marker genes.

### Histological and immunohistochemical staining

Skin equivalents and biopsies of preputial skin were fixed with 4% PFA at 4 °C overnight, processed for paraffin embedding using a Microm STP 120 tissue processor (Thermo Fisher Scientific), and 5 µm thick sections were prepared using a SM2010 R microtome (Leica). Tissue sections were deparaffinized in xylene and rehydrated using a descending ethanol series. For histological staining, sections were stained with hematoxylin and eosin (H&E) and mounted in Entellan. For immunofluorescence staining, antigens were retrieved by incubation in 10 mM sodium citrate, pH = 6 at 95 °C for 20 min. Sections were permeabilized in PBS- containing 0.3% Triton X-100 for 5 min and subsequently blocked with PBS-B for 30 min at RT. Next, sections were incubated overnight at 4 °C in a humidity chamber with the appropriate primary antibody (Anti-Collagen IV (ab6586; Abcam), anti-Cytokeratin 5 (MA5-12596; Thermo Fisher Scientific), anti-Cytokeratin 10 (M7002; DakoCytomation), anti-Cytokeratin 14 (HPA023040; Sigma-Aldrich), anti-E-Cadherin (610181; BD Biosciences), anti-Filaggrin (ab81468; Abcam), anti-Involucrin (MA5-11803; Thermo Fisher Scientific), anti-Loricrin (ab24722; Abcam), and anti-Vimentin (ab92547; Abcam)) at a dilution of 1:250, followed by washing three times for 5 min with PBS- containing 0.5% Tween-20 (PBS-T). Tissue sections were incubated with the appropriate secondary antibody

(anti-mouse IgG Alexa 555 (A31570; Thermo Fisher Scientific), anti-rabbit IgG Alexa 647 (A31573; Thermo Fisher Scientific)) at a dilution of 1:400 for 1 h at RT in the dark. After three washing steps with PBS-T for 5 min, sections were mounted in Fluoromount-G with DAPI. All images were acquired with a BZ-9000 fluorescence microscope (Keyence). The thickness of the cellular epidermis was measured with ImageJ/Fiji in H&E-stained sections of skin equivalents.

### Rheology

Mechanical properties of high- and low-density dermal equivalents as well as human skin biopsies were analyzed with an MCR 102 rheometer (Anton Paar). Pieces with a diameter of 8 mm were punched out of individual dermal equivalents and human skin biopsies (including dermis and epidermis) using an 8 mm tissue punch. Measurements were performed using a plate-plate geometry of 8 mm diameter and a gap size of 1 mm at RT. Amplitude sweeps from 0.01% to 100% strain deformation at 1 Hz were performed to investigate viscoelastic properties.

### Quantification and statistical analysis

The number of individual experiments is annotated in corresponding figure legends. Statistical significance was determined using the unpaired Student's t-test or Mann-Whitney U test in GraphPad Prism version 7 (GraphPad). P values correlate with symbols as follows: ns = not significant, $p > 0.05$, $^{*}p \leq 0.05$, $^{**}p \leq 0.01$, $***$ $p \leq 0.001$, $^{****}p \leq 0.0001$.

### Reporting summary

Further information on research design is available in the Nature Portfolio Reporting Summary linked to this article.

## Data availability

The sequencing data generated during this study has been deposited in the NCBI Gene Expression Omnibus (GEO) database under accession number GSE174198. Additional data regarding the sequencing analysis can be obtained from the Supplementary Data files. The human reference genome GRCh38, the *Trypanosoma brucei* TREU927 (version 48) genome as well as the VSGnome of AnTat 1.1E (EATRO1125) can be accessed through the included hyperlinks. All other data sets are available in the Source data file. Source data are provided with this paper.

## Code availability

The code used to generate the results presented in this publication can be downloaded from GitHub.

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

## Acknowledgements

The authors thank Panagiota Arampatzi, Nina DiFabion, Tobias Krammer, Christophe Toussaint, and Oliver Dietrich for guidance and useful discussions regarding scRNAseq; Stephan Löwe, Elisabeth Meyer-Natus, Tim Krüger, and Kathrin Weißenberg for technical assistance; Brooke Morriswood for critical reading of the manuscript. M.E. acknowledges funding from the following sources: German Research Foundation grant GRK 2157, German Research Foundation grant EN305, German Research Foundation grant SPP1726, German Research Foundation grant SPP2332, German-Israeli Foundation for Scientific Research and Development grant I-473-416.13/2018, European Union Innovative Training Network "Physics of Microbial Motility" grant No 955910, German Federal Ministry of Education and Research, Network of University Medicine (NUM) "Organo-Strat". A.E.S. and F.I. thank the Single Cell Center Würzburg for support. A.E.S. thanks the DFG GRK2157 and SFB DECIDE (Z project).

## Author contributions

Conceptualization, C.R., L.H., F.G.-B., and M.E.; Methodology, C.R., L.H., F.I., P.F., N.J., A.-E.S., M.E. and T.F.; Formal analysis, C.R., L.H., R.S., L.B., and E.V.; Investigation, C.R., L.H., F.I, N.G.J., M.E., and P.F.; Writing - Original Draft, C.R. and M.E.; Writing—review & editing, C.R., L.H., and M.E.; Supervision, H.W., L.B., A.-E.S., F.G.-B., and M.E.; Funding acquisition, M.E.

## Funding

## Competing interests

The authors declare no competing interests.
