## [Peer Review File · Nature Communications]

Vector-borne Trypanosoma brucei parasites develop in artificial human skin and persist as skin tissue formsREVIEWER COMMENTS

Reviewer #1 (Remarks to the Author):

The manuscript by Reuter et al, with the title "Vector-borne Trypanosoma brucei parasites develop in artificial human skin and persist as skin tissue forms" describes a new assay based human skin equivalent that after tse-tse fly infection allows detection of quiescent non-replicative parasites that could act as a reservoir of the parasite and explain persistent infection. The authors follow the change from metacyclic parasites inoculated by the fly to replicating skin-resident parasites within 24h, which later develop into quiescent parasites in skin equivalents. Moreover, they use single-cell RNA parasite sequencing to characterize the quiescent parasites, as well as size and mobility to confirm this novel skin stage of the parasite. This methodology could be used for studying other protozoan parasites as Leishmania.

The work is significant to the field because it doesn't need experimental infection in mice and the infection is performed directly in human cells. Sometimes good results in mice are not reproducible in humans. Using the skin equivalent the probability to get results applicable to humans is increased.

Although the authors get very nice results, since the immune response does not take place in the skin equivalent, I would say that the results do not fully support the conclusions. In particular, the one that claims that the skin is a reservoir of the parasite. Thus, to reach that conclusion an assay mimicking the immune system, using immune mediators, should be performed. Only then the skin equivalent would be useful for instance for drug screening.

In addition, some questions need clarification.

Major concerns

1. Please, extend the abstract including information about the most relevant techniques used to reach the conclusions.
2. Please explain in more detail experiments performed in references 6 and 7, were parasites from the original site of infection in the skin of mice?
3. Skin equivalents were produced that allows reproducing vectorial infection. Considering that in natural infection components of the saliva of the tse-tse fly enhance the infection by reducing inflammatory cytokines as IL-6 and TNF. Would the infection work using artificial inoculation, this may allow you to plan drug screening...
4. Quiescent parasites develop after conversion of metacyclic (MCF) to replicating skin-resident parasites. Is it possible that parasites lack some nutrients that cause quiescence? Has this been tested in vitro by nutrient depletion?
5. You show pictures of MCFs, skin-resident parasites at 12 h and 24 h, and culture BSF, however, you do not show the stumpy forms, do they look different than the quiescent ones?.
6. The skin equivalent lacks immune cells. One question that could be raised is whether quiescent parasites would resist the aggression of the immune mediators. Please discuss which mediators could be tested before transferring the parasites to the culture medium to check their viability and replicative capacities. As mentioned this could be the basis for testing drugs on quiescent/dormant parasites.
7. You suggest including macrophages for Leishmania parasites in the artificial-skin model Do you plan to include immune cells in your T. brucei artificial-skin model? Does it make sense in this case?. Please discuss.
8. In other protozoan parasites as Plasmodium and T. cruzi dormant parasites were described in target tissues. Why did you choose skin? You suggest using fat tissue, Why? Should quiescent/dormant parasites be studied in nerve cells as the main target of the infection?. Please discuss whether this phenomenon is exclusive of the skin...
9. If parasite persistence depends on quiescent skin resident parasites, how does this combines with antigenic variation, one of the hallmarks of sleeping sickness?.

Minor concerns

1. In figure 3B, if I understood correctly you took the forward-scatter width (FCS-W) of the parasites, but MFI stands in the axis label. Should you delete it?
2. In figure 4B, please indicate the percentages of variance explained in the PCA analysis.
3. In figure 4G, please indicate in the figure legend the purpose of highlighting the terms in red.

Reviewer #2 (Remarks to the Author):

The African trypanosome infects mammals through the bite of a tsetse fly. African trypanosomes appear to exploit the very different environmental niches of the mammalian body, by developing into metabolically discrete life cycle stages in different tissues. This manuscript from the Engstler lab develops a novel artificial skin model system, which they infect with African trypanosomes using tsetse flies. This is a very innovative approach to study a question, which has previously been considered to be experimentally intractable. Recently, "skin trypanosomes" have been discovered which appear to persist in asymptomatic patients. These quiescent trypanosomes could form an important reservoir for the disease in people who would previously not have been considered to have trypanosomiasis. However, very little is known about these intriguing "skin trypanosomes".

This study presents extensive validation of a novel artificial skin system, and argues that it is a good substitute for the real thing. There is a very good attempt at replicating a natural infection. Although one could possibly argue that three stacked artificial skin discs might not be the same as real skin, the use of tsetse flies for the infections is an excellent idea. Subsequently, the authors use a combination of scRNA-seq and microscopy to show that these skin resident trypanosomes have unique gene expression profiles, and are morphologically distinct from metacyclic form (MCF) cells.

The conclusion that these skin tissue forms (STFs) are slow replicating trypanosomes, which are maintained in this metabolic state through the environment of the 'artificial' skin in which they live appears to be well supported. The manuscript describes quite an impressive experimental tour de force, and in general the data presented look very nice.

However, there are a few issues that need to be resolved.

1. It was not clear from this manuscript if these STF trypanosomes can be considered a hybrid form between MCF and bloodstream form (BSF) trypanosomes. Alternatively, are the skin trypanosomes a novel type of BSF? The RNAseq analyses shown compare the STF transcriptomes with MSF transcriptomes, and show that there are clear differences. However, the authors also need to include BSF transcriptomes from this parasite strain in their RNAseq comparisons, as an additional important reference point. How similar are the transcriptomes of these skin trypanosomes with those from either MCF or BSF? In this vein, it is a shame that there don't appear to be transcriptomes of cells which have undergone the skin to medium transition in Figure 5, as these are presumably quite BSF-like.

2. The differences between the different transcriptomes (differentially upregulated genes) are shown in heat maps of selected upregulated transcripts (Figure 5). However, is it additionally possible to compare the entire transcriptomes with each other in volcano plots (for example in the Supplemental material)? This would allow the reader to put these results in the context of the entire transcriptomes of the different types (MCF, STF and BSF).

3. Are these STFs expressing MCF-VSGs or BSF VSGs? The STFs in the skin do not appear to be expressing AnTat1.1 to any significant extent. However, I think that there needs to be a more comprehensive analysis of the VSGs expressed by these STFs. Although the authors appear to be working with an unsequenced trypanosome strain, RNAseq analysis of populations of MCF trypanosomes from this *T. brucei* strain would give the authors a list of VSGs which must be the MCF-VSGs. What percentage of the STFs express mVSGs? If all of the VSGs expressed by the STFs are different from the documented mVSGs, this would argue that they are in the BSF VSG repertoire. How many different VSGs of these two different VSG classes are expressed by these STFs? This data needs to be shown (perhaps in the supplemental material), as it helps with the classification of this novel trypanosome type.

4. In Figure 3 it is argued that after increased time after infection, the MCFs resemble BSF cells in terms of the cell cycle, cell size, cell morphology and cell speed. However, a BSF control needs to be added to panels 3A and 3D.

5. I don't think that the FSC data in Fig. 3B is the appropriate way to show that the different types of trypanosomes are different sizes. Determining the forward scatter of elongated cells passing the laser beam in different orientations is very crude, even if the numbers analysed encompass all of the different possible orientations. The authors should instead determine the volume of these STF cells using a method determining the resistance of these cells in an electrical field. This would allow them to plot the volumes of these different cell types as pseudo-spheres. This is the methodology used by Coulter counters or CASY counters, and is far more accurate.

6. In supplementary figure S7A/B the authors show that a low number of stumpy form cells are present in the skin and that skin-derived parasites are able to infect tsetse flies from 4 dpi with low efficiency. Do the authors think that the low abundance stumpy form cells are allowing tsetse fly infection rather than the STFs? This was not very clear in the main text and should be elaborated on.

7. Is the artificial skin environment conducive to long term growth of rapidly dividing BSF trypanosomes? What happens if proliferative BSF cells are injected into it? This is of course very artificial, but do these BSF cells grow rapidly and then die, or do they develop into a slowly replicating BSF form?

8. Is there any evidence that the quiescent STF trypanosomes migrate out of the artificial skin disc into surrounding culture medium at a low rate, as could be presumed to happen in a natural infection in the host?

Minor comments:

-The rationale for using three skin discs stacked on top of each other could be explained in the text.

-In figure S1D the storage/loss modulus of the skin equivalents was calculated. How do these measurements compare with actual human skin?

-Figure 2B: how exactly are the trypanosomes fluorescently labelled? This was not clear from the figure legend.

-For flow cytometry of trypanosomes isolated from the skin how many events were recorded per sample? Given that there were on average 4000 MCFs injected per skin equivalent it would be good to know the average number of events per sample.

-Figure 5C shows the percentage of cells with "low protein synthesis". What exactly do the authors define as low protein synthesis?

-Title of Fig S8; "fluorescent" should be "fluorescence".

-Cyan means blue. In the heat map legends (i.e. Figure 5), do the authors want to say purple to yellow rather than purple to cyan?

Reviewer #3 (Remarks to the Author):

This manuscript describes the 'making' of a primary human skin equivalent and its use to study in detail the development of inoculated trypanosome parasites after the bite of an infected tsetse fly. This is an interesting approach opening the avenue of studying in a controlled in vitro primary skin environment the development of an inoculated pathogen. The detailed and timely follow up of the trypanosome at the population and individual level by scRNAseq, different microscopical imaging techniques and motility tracking give valuable and novel insights in the first developmental events when metacyclic trypanosomes switch host when inoculated by the infected tsetse into the mammalian host skin. The use of the cultivated primary skin equivalent certainly offers some

advantages since the experimental conditions are more simple and more controllable instead of using a 'live' mammalian host skin e.g. of lab mice. Moreover, a human skin model can be used this way. These are all noteworthy results and of high significance in the important research field of e.g. the vector-borne pathogens-skin interplay.

However, the presented in vitro primary skin equivalent remains a highly artificial system with a limited natural complexity and embedded in an artificial in vitro culture medium. Although the authors bring forward experimental data to demonstrate its similarity with the natural skin, the presented analysis/interpretation of the data remains limited and is strongly selected to favor the artificial skin model avoiding reflections about possible relevant differences that could impact the biological relevance of the in vitro primary skin model and the experimental observations. Indeed, the presented primary skin model harbours well-structured key cells that express significant differentiation markers but it remains a highly 'simplified' environment that is far from mimicking the complex dermal environment to which the parasites are exposed to when inoculated in the 'real' host skin, where e.g. the immediate and complex innate immune response generated by a various set of different cells (resident and bite site-recruited) play a determining role in the modulation of the bite site environment. Therefore, to make a case for the usefulness and biological relevance of the proposed experimental primary skin equivalent, the manuscript needs a more thorough, more detailed and unbiased analysis/reflection of as many variables of importance and with a good sense of criticism to compare the high density primary skin equivalent with the complexity of native human skin and its physiological /immunological environment. The authors present flow cytometry data, scRNA analysis, microscopical imaging data but only highlights the resemblance with the native human skin but what are the major differences, if any? This should be presented and discussed as well. This is essential as it will allow to estimate how relevant/close the in vitro primary skin model is (and the related experimental observations) to the natural skin situation.

- scRNAseq analysis on primary human skin equivalent: see fig S3; the epidermis and dermis were separated before performing scRNAseq (A); however, it is not clear how the results are presented (B and C) = not separately for epidermis/dermis?

- Fig.1 D to F: authors state that the 7 clusters with distinct expression profiles is in good agreement with the native human skin while referring to the Wang et al (2020) paper. If I'm correct, the latter only concerns a detailed scRNA seq analysis of the human skin epidermis, not including the dermis. So what about the comparison with the dermal gene expression profile in the native human skin? The dermal site is the most relevant for the inoculated trypanosomes.

- Page 5: low protein synthesis in MCF's in the skin at 5 and 7 dpi , downregulation of replication-associated genes, lower metabolic rate; based on this, the authors state that this indicates that the parasites go into a reversible quiescence program in the skin. However, could this overall lower cellular activity, after an initial period of strong proliferation in the skin, not simply be the result of a local deprivation of the available nutrients, slowing down the nutritional uptake and subsequent growth/metabolic rate of the clustered trypanosomes (pockets of trypanosomes entangled by collagen fibers)? How efficient is the replenishment of nutrients/medium within this high density artificial skin structure to reach the clustered parasites? The fact that these parasites immediately restart proliferation/protein synthesis once they are freed from this entanglement and are put in fresh culture medium, could indicate that the observed low overall parasite cellular is simply due to a local nutritional deprivation. In my view, the interpretation that parasites go into a reversible quiescent stage is not sufficiently evidenced and needs additional evidence.

- Page 5./ ... trypanosomes were found in the skin that expressed VSG AnTat1.1, an isoform characteristic of BSFs. This formulation needs more clarity and re-formulation. In fact, the AnTat 1.1 VSG is indeed a bloodstream VSG that is normally dominantly expressed during the first blood stream parasitemic wave of the specific strain that is used in this study, the *T. b. brucei* AnTat1.1 clone.

- After natural infection, skin equivalents were cultured for another 4 h, 12 h, 24 h, and 7 d respectively. In addition, freshly collected MCFs obtained from tsetse flies were used. How were the MCFs collected? Were they incubated in the same medium as the parasites from the

skin equivalents and were they processed (undergoing the same steps/treatments) for sorting as described for the parasites from the skin equivalents? If not, how can the authors exclude that some transcriptional changes compared with the MCFs are not linked to the isolation procedure /treatment prior to cell sorting.

- Fig.4C is difficult to read correctly: from this heatmap figure it seems that the MCF have quite a lot of upregulated genes which seems contradictory to their quiescent status. In fact, the informative value of this figure is not clear.

Fig S6. BARP genes were used for normalization as it is stated by the authors that BARP-genes are specifically expressed in epimastigote forms; based on this they excluded three such parasites from the MCF transcriptomes. However, it was recently demonstrated that barp transcripts are still abundantly present in T.b.brucei MCFs (see Casas-Sánchez et al., preprint: <https://doi.org/10.1101/477737>) so the low barp expression seems not an appropriate 'selection marker' for normalization towards MCFs.

Fig S6/ scRNA sequencing on MCF and artificial skin-inoculated parasites at different time points: for each different cell population around 25 single-cell transcriptomes were analyzed (for MCF: 48). I fully understand that it is costly and labor intensive but is the number of individual transcriptomes/cell population not too low to obtain robust data allowing reliable interpretation. Please comment on this, adding some relevant references that justifies this limited use of single cell transcriptomes.

Page 5: The authors write that the GO-analysis of differently-expressed genes revealed 'typical signatures' associated with each timepoint (Fig. 4 G, fig. S6 E, and data S4). This is vague; it is not clear what the authors mean here with these 'typical signatures'. From the data in the figures and excel this is not clear. This requires a better annotation/clarification of these 'typical signatures'.

Page 8/discussion: The authors state that they have clearly showed that the natural host environment is required for parasite development. I do not think it is appropriate to claim the merit for this. It has already been shown since many decades ago that human African trypanosomes have a complex life cycle switching between tsetse and the mammalian host, and that the natural host environment (in this case the skin where tsetse inoculates the parasites) is key for the parasite development. Maybe the authors wanted to express something else so they should re-phrase the sentence?

Page8/discussion: The authors should better link their findings with similar observations that were already reported in Caljon et al. (Plos Pathogens 2016) using a natural skin infection through tsetse bite. Indeed, the establishment of a local, dermal proliferative stage and the close interaction /entanglement of morphologically blood stream-like trypanosomes with the dermal collagen fibers/extracellular matrix were already reported in Caljon et al. and this is now confirmed in this study using the artificial skin model. This indicates that at least to some extent, trypanosomes are behaving similarly in the artificial skin model.

Page 9: The VSG repertoire of the pleiomorphic strain EATRO 1125 used in the study is not known. This statement is not entirely correct. The Tbb. AnTat 1.1 that is used in this study is a VSG Antat 1.1-selected clone derived from the T.b.b.AnTAR1 strain (= Antwerp Antigenic Repertoire 1 which includes the expression of different AnTat1.x VSGs during parasitemia waving in the host blood stream). This Tbb AnTAR1 is a cloned strain derived from the T. b. brucei isolate EATRO 1125.

Page 9-line 17: ... we clearly show the skin environment strongly influences the parasites. please correct to ... the artificial skin environment....

Reviewer #4 (Remarks to the Author):

Interesting and well-written manuscript and a very innovative application of a 3D skin model. There are still some questions regarding the skin model applied:

Why was the described compression model used, were collagen based 3D models without compression also used in comparison? How does the mechanical compression influence the gene expression of the skin model? Why was not a more stable alternative dermal matrix applied instead?

Modern 3D skin models try to reflect human skin in as many aspects as possible. Immune cells, such as macrophages, dendritic cells, or vascular cells, among others, are incorporated here to allow cross-talk with dermal fibroblasts and epidermal keratinocytes. Why was a model used here that only contains keratinocytes and fibroblasts?

The disadvantages of an in vitro skin model compared to the in vivo situation/animal model should also be discussed.

Vector-borne *Trypanosoma brucei* parasites develop in artificial human skin and persist as skin tissue forms

Response to reviewer's comments

We are grateful to the editors and reviewers for their thorough review, which has significantly improved our manuscript. We have successfully addressed all the raised queries and carefully considered the suggestions provided. However, we would like to emphasize that our manuscript contains an abundance of high-quality data, resulting in some good ideas being beyond the scope of our current study.

The proposal to incorporate an immune system into the skin model is undoubtedly a logical next step that we are actively pursuing. Nevertheless, it is crucial to acknowledge that integrating a skin-specific immune system or vascularization into the model is an extremely complex endeavor, considering the years of development and qualification invested in establishing the full-thickness model. In fact, one of the key strengths of our approach, differentiating it from animal models, is its modular nature. In the future, the skin model will exhibit increasing levels of complexity, enabling us to ascertain the impact of specific tissue components on infection progression.

Our current study already serves as an exemplary illustration of trypanosomes undergoing a transition to a previously overlooked life stage within the skin model, as confirmed through comprehensive metabolic and dual-single-cell RNA sequencing analyses. Intriguingly, this development did not appear to require the presence of an immune system. Additionally, our work represents the first successful combination of a vector and tissue model to achieve near-natural infection. Rather than employing monomorphic laboratory strains of the parasites, we cultivated pleomorphic trypanosomes, which pose considerable challenges *in vitro*. However, this cultivation approach ensures the maximum reproducibility of our system. The conventional method would have involved growing the parasites in mice.

In summary, we present a novel tissue model, the establishment of infection through the tsetse fly vector, and time-resolved single-cell analyses of pleomorphic parasites and skin cells. This unique combination of methodologies sets the stage for further advancements in understanding the influence of specific components within the tissue on infection dynamics.

Reviewer #1 (Remarks to the Author)

The manuscript by Reuter et al, with the title "Vector-borne *Trypanosoma brucei* parasites develop in artificial human skin and persist as skin tissue forms" describes a new assay based human skin equivalent that after tsetse fly infection allows detection of quiescent non-replicative parasites that could act as a reservoir of the parasite and explain persistent infection. The authors follow the change from metacyclic parasites inoculated by the fly to replicating skin-resident parasites within 24h, which later develop into quiescent parasites in skin equivalents. Moreover, they use single-cell RNA parasite sequencing to characterize the quiescent parasites, as well as size and mobility to confirm this novel skin stage of the parasite. This methodology could be used for studying other protozoan parasites as *Leishmania*. The work is significant to the field because it doesn't need experimental infection in mice and the infection is performed directly in human cells. Sometimes good results in mice are not reproducible in humans. Using the skin equivalent the probability to get results applicable to humans is increased.

Although the authors get very nice results, since the immune response does not take place in the skin equivalent, I would say that the results do not fully support the conclusions. In particular, the one that claims that the skin is a reservoir of the parasite. Thus, to reach that conclusion an assay mimicking the immune system, using immune mediators, should be performed. Only then the skin equivalent would be useful for instance for drug screening. In addition, some questions need clarification.

Answer: Our objective was to establish a reliable and replicable platform for vector-mediated infection of artificial skin. The notion that skin serves as a reservoir for trypanosomes dates back to the 1970s, and recent studies have revisited and substantiated this claim (see below). We are currently in the process of incorporating immune cells into our model, but this is a complex undertaking that requires several optimization steps and a plethora of controls. Therefore, we have decided not to include preliminary experiments with questionable validity. Additionally, the current skin model was not designed for drug screening purposes. However, this could be a future avenue of research, especially with regard to neglected tropical skin diseases, as the WHO recently highlighted.

Major concerns:

1. Please, extend the abstract including information about the most relevant techniques used to reach the conclusions.

Answer: The summary is unfortunately very limited in terms of word count. We tried to summarise the most important results and conclusions. Unfortunately, this did not leave much room for the methodology. Nevertheless, we have now added "by single-cell RNA sequencing" to address a key technique in the summary.

"We detailed the chronological order of the parasites' development in the skin by single-cell RNA sequencing, and found a rapid activation of metacyclic trypanosomes and differentiation to proliferative parasites."

2. Please explain in more detail experiments performed in references 6 and 7, were parasites from the original site of infection in the skin of mice?

Answer: The main text has to be very concise, so we are not able to give detailed information about the experiments mentioned in cited publications. Nevertheless, we have summarised the most important experiments from references 6 and 7. This information could be added as supplementary material, however, this an editorial decision.

Ref. 6: G. Caljon et al., The Dermis as a Delivery Site of Trypanosoma brucei for Tsetse Flies. PLoS Pathog 12, e1005744 (2016):

In this study, the authors established an experimental transmission model using fluorescently labeled parasites in mice. This model enabled them to investigate the fate of the parasites following natural transmission through a tsetse fly bite. The findings revealed that the parasites migrated from the site of dermal inoculation, leading to detectable parasite levels in the draining lymph nodes within 18 hours, and in the peripheral blood within 42 hours. This was demonstrated through the use of flow cytometry and RT-qPCR. Furthermore, by monitoring the progression of parasitemia in the peripheral blood of mice inoculated intradermally with varying numbers of metacyclic parasites extracted from tsetse flies, as well as purified bloodstream forms, the authors made an intriguing discovery. They found that metacyclic trypanosomes, in stark contrast to bloodstream form trypanosomes, exhibited high infectivity through the intradermal route. This highlights a significant

difference in their mode of transmission. Additionally, the authors suggested that a portion of the parasites remained in the skin and actively multiplied near the initial inoculation site within 18 hours post-infection. This indicates that the skin serves as a reservoir for the parasites, facilitating their proliferation.

Ref. 7: P. Capewell et al., The skin is a significant but overlooked anatomical reservoir for vector-borne African trypanosomes. Elife 5, (2016).

This study revealed the presence of significant quantities of trypanosomes within the skin following experimental infection, which could be transmitted to the tsetse vector, even in the absence of detectable parasitemia. Importantly, the study also demonstrated the existence of extravascular parasites in skin biopsies taken from undiagnosed individuals.

Mice were infected either through intra-peritoneal injection with bloodstream form parasites or via a single infective bite from a tsetse fly. Subsequently, the authors investigated the invasion of parasites into the skin tissue from the bloodstream.

3. Skin equivalents were produced that allows reproducing vectorial infection. Considering that in natural infection components of the saliva of the tse-tse fly enhance the infection by reducing inflammatory cytokines as IL-6 and TNF. Would the infection work using artificial inoculation, this may allow you to plan drug screening...

Answer: We have now done this experiment and can clearly state the tsetse saliva is not required for successful infection. The data will be published separately.

4. Quiescent parasites develop after conversion of metacyclic (MCF) to replicating skin-resident parasites. Is it possible that parasites lack some nutrients that cause quiescence? Has this been tested in vitro by nutrient depletion?

Answer: This is a good point. Based on the following facts and further experiments, we can rule out the possibility that nutrient deficiency causes dormancy:

(1) The skin cells would immediately react to nutrient deficiency. Since the skin equivalents are only supplied with nutrients from the basal side, the epidermis would first suffer from nutrient

deficiency and show poorly differentiated epidermal cell layers. However, this is not the case, as histology shows.

(2) Mammalian stage trypanosomes rely on glucose as their main source of energy. The diffusion of such a small substance in the interstitium of our skin models is so fast that even local depletion is physically impossible.

(3) In our experimental setup, we provided three infected skin equivalents with 7 ml of fresh medium every two days, ensuring that glucose deprivation is excluded. Typically, one uninfected skin equivalent is maintained with 1.75 ml of medium. As a result, there is an additional 1.75 ml available for infected skin equivalents to feed approximately 4×10^5 trypanosomes across the three skin equivalents (5.7×10^4 cells/ml). This quantity of medium is more than sufficient, especially considering that trypanosomes can easily reach cell densities of 10^6 cells/ml.

(4) We have now performed additional experiments with highly proliferative monomorphic bloodstream stage trypanosomes. These parasites can grow to very high cell numbers if sufficient nutrients are present. After injection into our skin models with a syringe, the trypanosomes reached numbers of up to 7×10^5 cells per model, equaling a density of 6×10^6 cells/ml. This shows that the availability of nutrients is not a limiting factor in our skin equivalents. We have added this new result to Figure 2D.

5. You show pictures of MCFs, skin-resident parasites at 12 h and 24 h, and culture BSF, however, you do not show the stumpy forms, do they look different than the quiescent ones?

Answer: Compared to BSFs or dormant STFs, stumpy-form parasites have a shorter, stout cell shape with a reduced flagellum length. Two images showing stumpy-form trypanosomes and skin-resident parasites 7 days post-infection have now been added to Fig. 3 C.

6. The skin equivalent lacks immune cells. One question that could be raised is whether quiescent parasites would resist the aggression of the immune mediators. Please discuss which mediators could be tested before transferring the parasites to the culture medium to check their viability and replicative capacities. As mentioned, this could be the basis for testing drugs on quiescent/dormant parasites.

Answer: This is a good point, but as mentioned earlier, the present work represents the first and most important step on our way to creating a modular and versatile skin tissue model for infection research. Due to the word count constraints of the manuscript, we regret that we cannot delve into the potential role of immune mediators. Nonetheless, we aim to address all the opportunities and challenges associated with skin models for infection research in an opinion article. Please find below an excerpt from the draft manuscript:

The initial cellular responders against injected trypanosomes in the skin are neutrophils, natural killer (NK) cells, and NK T-cells (Stijlemans et al., 2016; Caljon et al., 2018). This primary immune response targets the metacyclic VSG coat and can result in a localized skin reaction known as a chancre, which typically develops a few days after infection. The intensity of the immune response and the size of the chancre are correlated with the number of deposited metacyclic trypanosomes (Barry and Emery, 1984). Additionally, studies have demonstrated the crucial role of CD4+ T cells in the formation of chancres (Naessens et al., 2003).

*In the case of *T. brucei*, it has been observed that neutrophils are recruited to the dermal bite site within 4.5 hours. However, these early neutrophils surprisingly do not seem to contribute to the elimination of trypanosomes in the skin. Instead, their exact role in supporting the establishment of parasite infection remains unknown (Caljon et al., 2018).*

*Macrophages are the predominant immune cell population in the skin, responsible for monitoring the skin microenvironment for signs of cellular stress, tissue damage, or infection (Murray and Wynn, 2011). Our understanding of the role of macrophages in African trypanosomiasis primarily stems from studies involving experimental intraperitoneal or intravenous infections in mice. Consequently, the specific functions of macrophages in the skin during trypanosome infection remain poorly understood. Trypanosomes activate resident macrophages in the skin through pathogen-associated molecular patterns (PAMPs), such as soluble VSG released by the parasites. This activation leads to the production of inducible nitric oxide synthase (iNOS) enzyme, which generates toxic nitric oxide (NO). Studies have demonstrated the involvement of NO in the innate defense against intradermally injected bloodstream forms of *T. congolense* (Wei et al., 2011; Lu et al., 2011).*

*Dendritic cells (DCs) residing in the skin encompass epidermal Langerhans cells, which sample and present antigens from the epidermis to stimulate the adaptive immune response, as well as dermal DCs (Haniffa et al., 2015; Kaplan, 2017). While their role in suppressing *Leishmania* infections by*

modulating the behavior of regulatory T cells has been established, their precise involvement in African trypanosomiasis remains to be elucidated (Kautz-Neu et al., 2011). Recent research is using spatial transcriptomics to further investigate the role of skin responses in chronic T. brucei infections.

(<https://www.biorxiv.org/content/10.1101/2023.03.01.530674v1>).

7. You suggest including macrophages for Leishmania parasites in the artificial-skin model. Do you plan to include immune cells in your T. brucei artificial-skin model? Does it make sense in this case? Please discuss.

Answer: An immune-competent skin model would prove invaluable in gaining a deeper understanding of the initial interaction between trypanosomes (and other parasites) and the innate immune system. We have done the first steps. We have also conducted promising preliminary trials with *Leishmania*. We are, however, still far from having established a standard operating protocol. In fact, the lack of reproducibility of complex models is one of the biggest problems of tissue engineering in infection biology.

8. In other protozoan parasites as Plasmodium and T. cruzi dormant parasites were described in target tissues. Why did you choose skin? You suggest using fat tissue, Why? Should quiescent/dormant parasites be studied in nerve cells as the main target of the infection? Please discuss whether this phenomenon is exclusive of the skin...

Answer: Unlike other parasitic protozoa such as *Leishmania* and *Plasmodium*, African trypanosomes are strictly extracellular parasites. This means through vivid motility in the body fluids they can basically access any extracellular space. We have summarized this in an opinion article (Krüger et. al., 2018). While the interaction between the infectious metacyclic stage of *Trypanosoma brucei* and the host skin has long been recognized, the underlying cellular and molecular mechanisms remain poorly defined. Consequently, only a limited number of studies have investigated the early events of naturally transmitted trypanosome infections. The precise timing and mechanisms governing parasite differentiation in the skin remain elusive, as does the characterization of proliferative trypanosomes residing in the skin. Our skin model offers a unique advantage as it is well-defined and accessible during the early stages of infection. The presence of dormant parasites

in the skin equivalents underscores the potential significance of the host skin as a target tissue. Similar prospects may exist for subcutaneous adipose tissue. There are a number of publications highlighting adipose tissue as a reservoir for trypanosomes (Trindade et al., 2016; Machado et al., 2021; Machado et al., 2022; Trindade et al., 2022). Although our experiments with adipose tissue models are quite advanced and very promising, the qualification of the subcutaneous fat model is not yet complete.

9. If parasite persistence depends on quiescent skin resident parasites, how does this combine with antigenic variation, one of the hallmarks of sleeping sickness?

Answer: This is an excellent question that poses a challenge to answer. We propose the following scenario: Within the STFs, it is probable that a subset of the population continues to express the metacyclic VSG, while the activation of different bloodstream form expression sites occurs in another subset. Our speculation is that this arrangement allows the parasites to allocate time for recombination of new VSG genes into silent expression sites. To support this assumption, we have conducted additional experiments. When MCF were inoculated into the skin models and tested for the presence of AnTat 1.1 VSG (a bloodstream form VSG) on day 4 and day 7, we discovered that less than 0.1% of the cells were positive for AnTat 1.1. However, when MCF were directly inoculated into the cell culture medium (HMI9) and incubated for 7 days, approximately 15-20% of all cells exhibited positivity for AnTat 1.1. This indicates that BSF expression site activation is fully achieved in culture, but not in STFs. Our scRNAseq analyses further demonstrate that even after one week in the tissue equivalent, many cells continue to express mVSGs. This finding aligns with previous research conducted on mice, suggesting that mVSGs are expressed for up to 6 days (Esser et al., 1982). We have now included this data in the new Figure 6.

Minor concerns:

1. In figure 3B, if I understood correctly you took the forward-scatter width (FCS-W) of the parasites, but MFI stands in the axis label. Should you delete it?

Answer: As per the request of another reviewer, we have completely redone the process of measuring cell sizes. For this, we have devised an optimized workflow utilizing a Beckman Coulter Multisizer 4e. The results are now shown in Figure 3B.

2. In figure 4B, please indicate the percentages of variance explained in the PCA analysis.

Answer: We have now included the percentages of variance.

3. In figure 4G, please indicate in the figure legend the purpose of highlighting the terms in red.

Answer: The aim was to highlight significant biological processes at the respective time points. However, as this classification is subjective, the highlighting was removed.

Reviewer #2 (Remarks to the Author)

The African trypanosome infects mammals through the bite of a tsetse fly. African trypanosomes appear to exploit the very different environmental niches of the mammalian body, by developing into metabolically discrete life cycle stages in different tissues. This manuscript from the Engstler lab develops a novel artificial skin model system, which they infect with African trypanosomes using tsetse flies. This is a very innovative approach to study a question, which has previously been considered to be experimentally intractable. Recently, “skin trypanosomes” have been discovered which appear to persist in asymptomatic patients. These quiescent trypanosomes could form an important reservoir for the disease in people who would previously not have been considered to have trypanosomiasis. However, very little is known about these intriguing “skin trypanosomes”.

This study presents extensive validation of a novel artificial skin system, and argues that it is a good substitute for the real thing. There is a very good attempt at replicating a natural infection. Although one could possibly argue that three stacked artificial skin discs might not be the same as real skin, the use of tsetse flies for the infections is an excellent idea.

Subsequently, the authors use a combination of scRNA-seq and microscopy to show that these skin resident trypanosomes have unique gene expression profiles, and are morphologically distinct from metacyclic form (MCF) cells.

The conclusion that these skin tissue forms (STFs) are slow replicating trypanosomes, which are maintained in this metabolic state through the environment of the 'artificial' skin in which they live appears to be well supported. The manuscript describes quite an impressive experimental tour de force, and in general the data presented look very nice.

However, there are a few issues that need to be resolved.

1. It was not clear from this manuscript if these STF trypanosomes can be considered a hybrid form between MCF and bloodstream form (BSF) trypanosomes. Alternatively, are the skin trypanosomes a novel type of BSF? The RNAseq analyses shown compare the STF transcriptomes with MSF transcriptomes, and show that there are clear differences. However, the authors also need to include BSF transcriptomes from this parasite strain in their RNAseq comparisons, as an additional important reference point. How similar are the transcriptomes of these skin trypanosomes with those from either MCF or BSF? In this vein, it is a shame that there don't appear to be transcriptomes of cells which have undergone the skin to medium transition in Figure 5, as these are presumably quite BSF-like.

Answer: We have now included RNAseq data from BSF parasites. We do not consider the STFs to be either specialized bloodstream forms or hybrids between metacyclic and bloodstream forms.

The STFs are no bloodstream forms for the following reasons: (i) they express mVSGs, (ii) they do not exhibit quorum sensing, (iii) they are not infective to tsetse flies, and (iv) they have a distinct gene expression profile. Similarly, the STFs cannot be classified as metacyclic forms because (i) they proliferate (ii) they differ in morphology and cell size and (iii) they display a unique gene expression profile. Our recent research has revealed an unexpected plasticity in the life cycle of trypanosomes, which is likely true for the life cycles of other parasites as well. We should anticipate that the life cycle stages we have studied in cell culture exhibit greater diversity in different habitats within the host and vector. However, the STFs possess all the characteristics necessary to be considered a distinct life cycle form, and not a hybrid stage. We agree that it would have been desirable to also have transcriptomes from trypanosomes that have undergone the transition from skin to medium.

Although this is outside the scope of the present study, the experiments will be performed and published separately.

2. The differences between the different transcriptomes (differentially upregulated genes) are shown in heat maps of selected upregulated transcripts (Figure 5). However, is it additionally possible to compare the entire transcriptomes with each other in volcano plots (for example in the Supplemental material)? This would allow the reader to put these results in the context of the entire transcriptomes of the different types (MCF, STF and BSF).

Answer: We agree that volcano plots are often the better option to visualize differential gene expression, especially when two data sets are compared to each other. Thus, we have exchanged the heat maps in Figure 5 with volcano plots and added another volcano plot for comparison of early vs. late time points to Figure S6 (S6 E). Additionally, we have added Figure S9 to the supplementary material, containing volcano plots showing the differential gene expression between important time points during parasite differentiation (MCF, 24 h, 7 d) and BSF as a known and characterized reference stage.

3. Are these STFs expressing MCF-VSGs or BSF VSGs? The STFs in the skin do not appear to be expressing AnTat1.1 to any significant extent. However, I think that there needs to be a more comprehensive analysis of the VSGs expressed by these STFs. Although the authors appear to be working with an unsequenced trypanosome strain, RNAseq analysis of populations of MCF trypanosomes from this *T. brucei* strain would give the authors a list of VSGs which must be the MCF-VSGs. What percentage of the STFs express mVSGs? If all of the VSGs expressed by the STFs are different from the documented mVSGs, this would argue that they are in the BSF VSG repertoire. How many different VSGs of these two different VSG classes are expressed by these STFs? This data needs to be shown (perhaps in the supplemental material), as it helps with the classification of this novel trypanosome type.

Answer: We have now included the VSG analysis in the new Figure 6. The data reveal the transition from sole expression of mVSGs to the expression of BSF VSGs. Interestingly, the STFs can express the mVSG even after 7 days. This agrees with in vivo observations in mice suggesting that mVSGs

are expressed for up to one week post parasite transmission to the mammalian host (Esser et al., 1982).

4. In Figure 3 it is argued that after increased time after infection, the MCFs resemble BSF cells in terms of the cell cycle, cell size, cell morphology and cell speed. However, a BSF control needs to be added to panels 3A and 3D.

Answer: We have generated and incorporated data on the cell cycle distribution of BSF parasites obtained through flow cytometry into Figure 3A. Additionally, we have included data on the average and maximum swimming speeds of BSF parasites within the skin equivalent. The parasites were injected using a syringe, and their swimming speeds were subsequently measured. We mentioned this in the text (p.5, lines 9 - 10).

5. I don't think that the FSC data in Fig. 3B is the appropriate way to show that the different types of trypanosomes are different sizes. Determining the forward scatter of elongated cells passing the laser beam in different orientations is very crude, even if the numbers analysed encompass all of the different possible orientations. The authors should instead determine the volume of these STF cells using a method determining the resistance of these cells in an electrical field. This would allow them to plot the volumes of these different cell types as pseudo-spheres. This is the methodology used by Coulter counters or CASY counters, and is far more accurate.

Answer: Initially, we had reservations about employing this methodology for trypanosomes due to their irregular cell shape, constantly undergoing deformation caused by the beating of the attached flagellum. Nonetheless, we done the experiment, and we found that it yielded highly accurate results. It is important to note that to fully utilize the technique for trypanosomes, several optimization steps had to be undertaken. But we succeeded and – in the end we bought the machine. Therefore, we would like to express our utmost gratitude to the reviewer for pushing us in the right direction. We present the newly obtained data in Figure 3B.

6. In supplementary figure S7A/B the authors show that a low number of stumpy form cells are present in the skin and that skin-derived parasites are able to infect tsetse flies from 4 dpi

with low efficiency. Do the authors think that the low abundance stumpy form cells are allowing tsetse fly infection rather than the STFs? This was not very clear in the main text and should be elaborated on.

Answer: This is a good point, we have included the information on page 11, lines 23-25:

“The hypothesis of specific skin tissue forms is further supported by the very low infection rates of tsetse flies that were fed with skin-derived trypanosomes. We have recently shown that a single bloodstream stage trypanosome suffices to successfully infect a tsetse fly, and that it does not matter if this is a proliferative BSF or stumpy stage parasite (40). Thus, if the STFs would be BSFs, we would expect much higher fly infection rates. This suggests that STFs are not tsetse-infective.”

Therefore, we hypothesize that tsetse flies were either infected by the low abundance stumpy forms or by still remaining bloodstream forms in the skin.

7. Is the artificial skin environment conducive to long term growth of rapidly dividing BSF trypanosomes? What happens if proliferative BSF cells are injected into it? This is of course very artificial, but do these BSF cells grow rapidly and then die, or do they develop into a slowly replicating BSF form?

Answer: We have now investigated this by syringe-injecting BSF cells of both, a monomorphic as well as a pleomorphic *T. brucei* line, into the skin models. While monomorphic BSF cells grew to much higher numbers than fly-transmitted parasites (Figure 2D), this is not the case for pleomorphic BSFs. In contrast, these cells differentiate to stumpy forms within the models without reaching cell densities required for quorum sensing-dependent differentiation in flask culture. After 4-5 days post injection, all injected pleomorphic BSFs had died off. It needs to be further investigated what causes this reaction, but it once again emphasizes the difference between BSFs and STFs, of which the latter do not become stumpy to a significant extent despite surpassing critical parasite densities inside the models.

Once completed, this data will be published separately.

8. Is there any evidence that the quiescent STF trypanosomes migrate out of the artificial skin

disc into surrounding culture medium at a low rate, as could be presumed to happen in a natural infection in the host?

Answer: To test the possibility that the skin-residing trypanosomes escape from the hdSEs into the surrounding culture medium the supernatant was examined for the presence of parasites. The culture medium contained on average less than 8,000 parasites on each of the three time intervals tested. In total, this adds up to about 12,000 parasites in one week. Moreover, it must be taken into account that three infected hdSEs were always cultured together. Thus, the trypanosomes detected in the supernatant originated from three hdSEs. Thus, with 3.4 % per week, parasite escape from the skin models is negligible. We have now added the data to Fig. S8.

Minor comments:

-The rationale for using three skin discs stacked on top of each other could be explained in the text.

Answer: We have added the explanation to the text:

Page 4, lines 7-15: *“To simulate the infection process in the most natural way, tsetse flies were employed to transmit the infective MCFs to the hdSEs (Fig. 2 A and Movie S1). To achieve this, we stacked three hdSEs on top of each other, resulting in a final height of 3 mm. The reason for stacking the hdSEs was due to the proboscis of the tsetse fly, which has a length of 2 mm. In contrast, the hdSEs have a standard height of only 1 mm. Without stacking them, the flies would easily bite through the hdSEs, leading to the majority of the parasites being deposited underneath. The analysis of the skin lesion (fig. S5 A) revealed a complex deposition of the parasites within an intricate bite path network in the dermis (Fig. 2 B).”*

-In figure S1D the storage/loss modulus of the skin equivalents was calculated. How do these measurements compare with actual human skin?

Answer: We have done the experiment and now include data on the mean storage modulus of skin biopsies (Fig. 1B and Fig. S1D). As expected, the mean storage modulus was about 3-times higher when compared to the high-density skin equivalents. This may in part be caused by the presence of

the epidermis in skin explants. The skin equivalents were measured without epidermis. In conclusion, our model also mechanically resembles native skin well.

-Figure 2B: how exactly are the trypanosomes fluorescently labelled? This was not clear from the figure legend.

Answer: The fluorescent protein tdTomato was used for fluorescent labelling of trypanosomes. The plasmid pTSARib_tdTomato is integrated into the ribosomal DNA locus under the control of the rDNA promoter (Xong et al., 1998). Thus, the fluorescent protein tdTomato is constitutively expressed in the cytoplasm of all life cycle stages. We have added the information to the legend of Fig. 2B:

“Stereo-fluorescence microscopy detected tsetse-transmitted metacyclic trypanosomes in the dermis of a skin equivalent immediately after infection. Parasites (yellow), constitutively expressing the fluorescent protein tdTomato from the rDNA promoter, were found in multiple finger-like lesions (white arrows), probably corresponding to the bite path of the tsetse fly. Scale bar, 500 μ m.”

-For flow cytometry of trypanosomes isolated from the skin how many events were recorded per sample? Given that there were on average 4000 MCFs injected per skin equivalent it would be good to know the average number of events per sample.

Answer: For each sample we recorded approximately 10,000 trypanosomes. On average, around 4,000 MCFs were injected per tsetse fly (Fig. S5 B). Considering that six tsetse flies were used to infect a single skin equivalent, it can be estimated that approximately 24,000 MCFs were present in each skin equivalent at time point t_0 .

We have now added this information to the methods section “flow cytometry”:

“10,000 trypanosomes were analyzed by gating first on the tdTomato signal to exclude NHDF and extracellular matrix proteins from analysis.”

We have also added the information to the legend of Fig. 5A:

“ 1×10^4 parasites were analyzed per condition.”

-Figure 5C shows the percentage of cells with “low protein synthesis”. What exactly do the authors define as low protein synthesis?

Answer: The lowest synthesis rate measured in cultured BSF served as threshold. Parasites that revealed fluorescence below this value, as indicated by the red line in Fig. 3 F, were classified as "low protein synthesis parasites". We have included this explanation in the figure legends.

Legend to Fig. 5C: *"Quantification of cells classified as "parasites with low protein synthesis", indicated by the red line in Fig. 3F, in BSFs, MCFs, skin-resident trypanosomes and parasites at 7 dpi transferred to culture medium for one day (skin-to-medium)".*

Legend to Fig. 3F: *"Quantification of protein synthesis rates of BSFs, MCFs, and skin-resident parasites at various timepoints post-infection. Skin-resident parasites were additionally transferred to culture medium for 24 h after 7 dpi (skin-to-medium). The fluorescence intensity of incorporated AHA in individual parasites was measured with ImageJ/Fiji and the corrected total cell fluorescence (CTCF) was calculated. Data are median \pm IQR (n = 3; 136 - 292 parasites were analyzed per timepoint). Mann-Whitney U-test, **** p < 0.0001, ns, not significant. The red line defines the threshold for classification as "parasites with low protein synthesis"."*

-Title of Fig S8; "fluorescent" should be "fluorescence".

Answer: We thank the reviewer for such precise reading. We have corrected the title.

-Cyan means blue. In the heat map legends (i.e. Figure 5), do the authors want to say purple to yellow rather than purple to cyan?

Answer: Again, we thank the reviewer. We have corrected the figure legend.

Reviewer #3 (Remarks to the Author)

This manuscript describes the 'making' of a primary human skin equivalent and its use to study in detail the development of inoculated trypanosome parasites after the bite of an infected tsetse fly. This is an interesting approach opening the avenue of studying in a controlled in vitro primary skin environment the development of an inoculated pathogen. The detailed and

timely follow up of the trypanosome at the population and individual level by scRNAseq, different microscopical imaging techniques and motility tracking give valuable and novel insights in the first developmental events when metacyclic trypanosomes switch host when inoculated by the infected tsetse into the mammalian host skin. The use of the cultivated primary skin equivalent certainly offers some advantages since the experimental conditions are more simple and more controllable instead of using a 'live' mammalian host skin e.g. of lab mice. Moreover, a human skin model can be used this way. These are all noteworthy results and of high significance in the important research field of e.g. the vector-borne pathogens-skin interplay.

However, the presented in vitro primary skin equivalent remains a highly artificial system with a limited natural complexity and embedded in an artificial in vitro culture medium. Although the authors bring forward experimental data to demonstrate its similarity with the natural skin, the presented analysis/interpretation of the data remains limited and is strongly selected to favor the artificial skin model avoiding reflections about possible relevant differences that could impact the biological relevance of the in vitro primary skin model and the experimental observations. Indeed, the presented primary skin model harbours well-structured key cells that express significant differentiation markers but it remains a highly 'simplified' environment that is far from mimicking the complex dermal environment to which the parasites are exposed to when inoculated in the 'real' host skin, where e.g. the immediate and complex innate immune response generated by a various set of different cells (resident and bite site-recruited) play a determining role in the modulation of the bite site environment. Therefore, to make a case for the usefulness and biological relevance of the proposed experimental primary skin equivalent, the manuscript needs a more thorough, more detailed and unbiased analysis/reflection of as many variables of importance and with a good sense of criticism to compare the high-density primary skin equivalent with the complexity of native human skin and its physiological /immunological environment.

1. The authors present flow cytometry data, scRNA analysis, microscopical imaging data but only highlights the resemblance with the native human skin but what are the major differences, if any? This should be presented and discussed as well. This is essential as it will allow to estimate how relevant/close the in vitro primary skin model is (and the related experimental observations) to the natural skin situation.

Answer: The reviewer understandably raises concerns about the validity of our skin model and how it compares to reality, which is **the** crucial question in tissue engineering and organoid research. Our approach was to start with a relatively simple but biologically and mechanically accurate model that can be consistently reproduced. One encouraging result was that this skin equivalent was accepted by the tsetse fly, and we observed that metacyclic trypanosomes, which had been arrested in the cell cycle, readily reentered the cell cycle within the skin model. Interestingly, the trypanosomes within the skin model went on to develop into a previously unknown life cycle stage, the skin tissue form. For triggering this developmental transition our skin model was required and sufficient. The presence of immune cells, neurons, or vasculature was not necessary. However, it remains uncertain whether their inclusion would alter the trypanosome's behavior. Hence, we are currently in the process of systematically introducing these components to gradually refine the skin model. The ability to work with modular systems is indeed a significant advantage of tissue models. We would like to add that a similarly controllable and reproducible approach with tsetse skin infections of laboratory animals is hardly imaginable, not to mention the differences of mouse and human skin or ethical concerns.

On page 13, lines 3 – 8 we now concisely discuss the opportunities and limitation of engineered tissues for infection research.

" However, it is important to acknowledge that these models do not possess the same level of complexity as native human skin in terms of immune competency, vascularization, and innervation. Furthermore, the in vitro models lack the systemic component. In contrast, in vivo animal models provide these essential components, albeit with species-specific variations. Therefore, for future studies, it is crucial to carefully consider whether an in vitro or in vivo model is more appropriate for the specific research question at hand."

2. scRNAseq analysis on primary human skin equivalent: see fig S3; the epidermis and dermis were separated before performing scRNAseq (A); however, it is not clear how the results are presented (B and C) = not separately for epidermis/dermis?

Answer: To isolate keratinocytes from the epidermis and fibroblasts from the dermis, the epidermis and dermis were first separated, followed by performing distinct tissue dissociation treatments. After successfully isolating viable keratinocytes and fibroblasts, the cells were sorted using FACS.

Subsequently, both cell types were combined, and the resulting cell suspension was utilized for scRNAseq. During the data analysis, distinct cell types and sub-populations were distinguished based on known marker genes (Fig. 1D, F, E). The quality control of the data analysis is presented in Fig. S3B. Additionally, the expression of extracellular matrix-associated genes for both cell types and sub-populations is depicted in Fig. S3C.

3. Fig.1 D to F: authors state that the 7 clusters with distinct expression profiles is in good agreement with the native human skin while referring to the Wang et al (2020) paper. If I'm correct, the latter only concerns a detailed scRNA seq analysis of the human skin epidermis, not including the dermis. So what about the comparison with the dermal gene expression profile in the native human skin? The dermal site is the most relevant for the inoculated trypanosomes.

Answer: We have now included another publication (Tabib et. al, 2018) that defines the major fibroblast populations in human dermis. SFRP2/DPP4 are expressed in our skin equivalents. We have included this sentence into the Supplementary text about "High-density skin equivalents recapitulate key anatomical, cellular, and functional aspects of native human skin" (Supplementary material, p 12):

„SFRP2 and DPP4 expression was detected in the FB cluster, characterizing the most abundant major fibroblast population within the dermis. Cells expressing these markers are known to be small and elongated and express high levels of extracellular matrix components¹⁹.”

4. Page 5: low protein synthesis in MCF's in the skin at 5 and 7 dpi, downregulation of replication-associated genes, lower metabolic rate; based on this, the authors state that this indicates that the parasites go into a reversible quiescence program in the skin. However, could this overall lower cellular activity, after an initial period of strong proliferation in the skin, not simply be the result of a local deprivation of the available nutrients, slowing down the nutritional uptake and subsequent growth/metabolic rate of the clustered trypanosomes (pockets of trypanosomes entangled by collagen fibers)? How efficient is the replenishment of nutrients/medium within this high-density artificial skin structure to reach the clustered parasites? The fact that these parasites immediately restart proliferation/protein synthesis

once they are freed from this entanglement and are put in fresh culture medium, could indicate that the observed low overall parasite cellular is simply due to a local nutritional deprivation. In my view, the interpretation that parasites go into a reversible quiescent stage is not sufficiently evidenced and needs additional evidence.

Answer: This is a good point. Based on the following facts and further experiments, we can rule out the possibility that nutrient deficiency causes dormancy:

(1) The skin cells would immediately react to nutrient deficiency. Since the skin equivalents are only supplied with nutrients from the basal side, the epidermis would first suffer from nutrient deficiency and show poorly differentiated epidermal cell layers. However, this is not the case, as histology shows.

(2) Mammalian stage trypanosomes rely on glucose as their main source of energy. The diffusion of such a small substance in the interstitium of our skin models is so fast that even local depletion is physically impossible.

(3) In our experimental setup, we provided three infected skin equivalents with 7 ml of fresh medium every two days, ensuring that glucose deprivation is excluded. Typically, one uninfected skin equivalent is maintained with 1.75 ml of medium. As a result, there is an additional 1.75 ml available for infected skin equivalents to feed approximately 4×10^5 trypanosomes across the three skin equivalents (5.7×10^4 cells/ml). This quantity of medium is more than sufficient, especially considering that trypanosomes can easily reach cell densities of 10^6 cells/ml.

(4) We have now performed additional experiments with highly proliferative monomorphic bloodstream stage trypanosomes. These parasites can grow to very high cell numbers if sufficient nutrients are present. After injection into our skin models with a syringe, the trypanosomes reached numbers of up to 7×10^5 cells per model, equaling a density of 6×10^6 cells/ml. This shows that the availability of nutrients is not a limiting factor in our skin equivalents. We have added this new result to Figure 2D.

5. Page 5./ ... trypanosomes were found in the skin that expressed VSG AnTat1.1, an isoform characteristic of BSFs. This formulation needs more clarity and re-formulation. In fact, the

AnTat 1.1 VSG is indeed a bloodstream VSG that is normally dominantly expressed during the first blood stream parasitemic wave of the specific strain that is used in this study, the T. b. brucei AnTat1.1 clone.

Answer: We have added this sentence to the discussion on page 10, lines 6 - 9:

“VSG AnTat 1.1 is an isoform that is commonly expressed in the AnTat1.1 clone of the pleomorphic strain EATRO 1125 during the first wave of blood stream parasitemia, and we indeed found this VSG to be among the expressed VSG mRNAs at day 7 dpi.”

6. After natural infection, skin equivalents were cultured for another 4 h, 12 h, 24 h, and 7 d respectively. In addition, freshly collected MCFs obtained from tsetse flies were used. How were the MCFs collected? Were they incubated in the same medium as the parasites from the skin equivalents and were they processed (undergoing the same steps/treatments) for sorting as described for the parasites from the skin equivalents? If not, how can the authors exclude that some transcriptional changes compared with the MCFs are not linked to the isolation procedure /treatment prior to cell sorting.

Answer: MCFs were obtained from tsetse flies through continuous salivation into cultivation medium, and subsequently incubated in the same medium as the trypanosomes isolated from the skin. To ensure consistency, the experimental procedures involved in sample preparation for RNA sequencing (centrifugation, cell sorting by FACS, and cell lysis) were performed on the MCFs following the same protocols used for the trypanosomes isolated from the skin.

To provide further clarification, we have rephrased the text in the section describing the method for single-cell RNA sequencing of trypanosomes.

“scRNAseq of trypanosomes: After natural infection, skin equivalents were cultured for another 4 h, 12 h, 24 h, and 7 d, respectively. In addition, freshly collected MCFs obtained from tsetse flies harvested by continuous salivation into medium and flask-cultured BSFs were used. To ensure consistency, the experimental procedures involved in sample preparation for RNA sequencing (centrifugation, cell sorting by FACS, and cell lysis) were performed on the MCFs and BSFs following the same protocols used for the trypanosomes isolated from the skin.”

7. Fig.4C is difficult to read correctly: from this heatmap figure it seems that the MCF have quite a lot of upregulated genes which seems contradictory to their quiescent status. In fact, the informative value of this figure is not clear.

Answer: A dormant cell expresses quiescence-specific genes. The expression of these genes may be absent or significantly lower in proliferating or differentiating cells. Consequently, during a differential gene expression analysis and comparison with other proliferating or differentiating cells, these genes appear upregulated.

8. Fig S6. BARP genes were used for normalization as it is stated by the authors that BARP-genes are specifically expressed in epimastigote forms; based on this they excluded three such parasites from the MCF transcriptomes. However, it was recently demonstrated that *barp* transcripts are still abundantly present in *T.b.brucei* MCFs (see Casas-Sánchez et al., preprint: <https://doi.org/10.1101/477737>) so the low *barp* expression seems not an appropriate 'selection marker' for normalization towards MCFs.

Answer: We have not normalized towards MCFs, but just excluded 3 out of 48 cells that expressed BARP. This decision was made based on two publications that demonstrate that BARP is not expressed in metacyclic trypanosomes: Vigneron et al. (<https://doi.org/10.1073/pnas.1914423117>) show that *barp* genes are expressed in epimastigotes but not in metacyclics. Howick et al. (<https://doi.org/10.1371/journal.ppat.1010346>) also contradict that *barp* genes are expressed in MCFs: "*Mature metacyclics can be unequivocally identified by their lack of expression of the cell surface proteins GPEET, EP and BARP, replaced by expression of a single mVSG gene in each cell.*"

9. Fig S6/ scRNA sequencing on MCF and artificial skin-inoculated parasites at different time points: for each different cell population around 25 single-cell transcriptomes were analyzed (for MCF: 48). I fully understand that it is costly and labor intensive but is the number of individual transcriptomes/cell population not too low to obtain robust data allowing reliable interpretation. Please comment on this, adding some relevant references that justifies this limited use of single cell transcriptomes.

Answer: We anticipated this question. It is worth noting that there are several previous reports that have utilized similarly low numbers of transcriptomes, and in some cases even fewer.

We have now included additional data and conducted extensive work using 10x sequencing, as it had been meanwhile published for *T. brucei*. We even acquired a 10x Chromium Controller. However, during our sequencing trials, we discovered that 10x multiplexing was not compatible with trypanosomes. We reached out to the authors of the *T. brucei* study, as it had not been mentioned in their publication. In this study the various stages were categorized solely based on bioinformatics analysis, which was not feasible for our more complex experiments. While SMARTseq is not a high-throughput method, it is robust, reliable, and generates much deeper sequence information compared to 10x. More recently, we successfully employed SMARTseq3 in collaboration with Nicolai Siegel from LMU Munich. Although revisiting the skin infection experiments is feasible, it would be highly time-consuming. Hence, we made the decision not to further delay the revision of our manuscript, which now includes an abundance of additional results. Instead, we have included the following disclaimer on page 9, lines 18 – 21 in our discussion: "*The number of SMARTseq transcriptomes presented is limited, and there is a possibility that we may have overlooked subtle differences between the cell populations. However, the aim of our experiments was to identify the major differences, which we have achieved.*" Further in-depth analyses will be the objective of a follow-up study.

10. Page 5: The authors write that the GO-analysis of differently-expressed genes revealed 'typical signatures' associated with each timepoint (Fig. 4 G, fig. S6 E, and data S4). This is vague; it is not clear what the authors mean here with these 'typical signatures'. From the data in the figures and excel this is not clear. This requires a better annotation/clarification of these 'typical signatures'.

Answer: We agree with the reviewer the phrase 'typical signatures' is useless. Indeed, one can question the extremely coarse instrument of GO analysis, but it is widely used and still the standard. Our objective was to convey that each time point (MCF, 4h, 12h, 24h, 7d) was associated with a distinct set of Gene Ontology terms. To enhance clarity and understanding, we have rephrased the sentence as follows: "*More globally, Gene Ontology (GO) analysis of differentially-expressed genes revealed that distinct sets of GO terms were associated with each timepoint (Fig. 4 G, fig. S6 E, and data S4).*"

11. Page 8/discussion: The authors state that they have clearly showed that the natural host environment is required for parasite development. I do not think it is appropriate to claim the merit for this. It has already been shown since many decades ago that human African trypanosomes have a complex life cycle switching between tsetse and the mammalian host, and that the natural host environment (in this case the skin where tsetse inoculates the parasites) is key for the parasite development. Maybe the authors wanted to express something else so they should re-phrase the sentence?

Answer: We wrote: *“We have established an advanced skin model and validated it through tsetse-borne infection with trypanosomes. We clearly show that the natural host environment is required for parasite development. Tsetse-transmitted cell cycle-arrested MCFs are rapidly activated in the artificial skin and establish a proliferative trypanosome population.”*

We express our gratitude to the reviewer and acknowledge that our previous statement in this specific context could lead to misunderstandings. Our data indeed indicate that the natural host environment is not essential for the successful differentiation of cell cycle-arrested MCFs into proliferative BSFs (Figure S8 B). Rather, our intended conclusion was that the development of quiescent skin tissue forms relies on the presence of the near natural host environment, as these forms were not observed in the culture medium alone. To clarify this, we have revised the sentence on page 9, lines 2 - 4 as follows:

“We have established an advanced skin model and validated it through tsetse-borne infection with trypanosomes. We clearly show that this near-natural host environment supports parasite development. Tsetse-transmitted cell cycle-arrested MCFs are rapidly activated in the artificial skin and establish a proliferative trypanosome population.”

12. Page8/discussion: The authors should better link their findings with similar observations that were already reported in Caljon et al. (Plos Pathogens 2016) using a natural skin infection through tsetse bite. Indeed, the establishment of a local, dermal proliferative stage and the close interaction /entanglement of morphologically blood stream-like trypanosomes with the dermal collagen fibers/extracellular matrix were already reported in Caljon et al. and this is now confirmed in this study using the artificial skin model. This indicates that at least to some extent, trypanosomes are behaving similarly in the artificial skin model.

Answer: On page 3, lines 1-2 we write: *”A subpopulation of tsetse-injected trypanosomes was found to reside and proliferate in the skin at the bite site (6).”* Our reference 6 is Caljon *et al.*, 2016. On page 9, lines 8-11 we write: *“We found that the parasites re-entered the cell cycle already between 6 and 12 hours. Moreover, protein synthesis is reactivated very rapidly, after just 1 hour, and peaks within 1 day. These findings are consistent with natural infections of mice showing that MCFs start multiplying in the skin within 18 hours of transmission (6).”*

With these statements, we have clearly described our observations on the course of parasite development in the context of the results of Caljon *et al.*.

13. Page 9: The VSG repertoire of the pleiomorphic strain EATRO 1125 used in the study is not known. This statement is not entirely correct. The Tbb. AnTat 1.1 that is used in this study is a VSG Antat 1.1-selected clone derived from the T.b.b.AnTAR1 strain (= Antwerp Antigenic Repertoire 1 which includes the expression of different AnTat1.x VSGs during parasitemia waving in the host blood stream). This Tbb AnTAR1 is a cloned strain derived from the T. b. brucei isolate EATRO 1125.

Answer: The VSG repertoire of the pleomorphic strain AnTat 1.1 used in this study is no longer unknown. In the course of revision we have collected data on VSG expression which is now included in Fig. 6. Furthermore, others also have added knowledge on the VSG repertoire. Data on metacyclic VSGs has been published in PLoS Pathogens and is cited in our manuscript. In addition, there is a preprint that adds more information on BSF VSGs.

14. Page 9-line 17: ... we clearly show the skin environment strongly influences the parasites. please correct to ... the artificial skin environment....

Answer: We agree with the reviewer and have reworded the sentence as follows:

“Although the reason(s) why trypanosomes persist in mammalian skin remain unknown (6, 7), we clearly show that the artificial skin environment strongly influences the parasites.”

Reviewer #4 (Remarks to the Author):

Interesting and well-written manuscript and a very innovative application of a 3D skin model. There are still some questions regarding the skin model applied:

1. Why was the described compression model used, were collagen based 3D models without compression also used in comparison?

Answer: Uncompressed collagen models have been extensively utilized for various applications in recent years. However, standardizing these models proves challenging due to the fibroblast-mediated contraction, resulting in varying sizes at the end of the culture period. Therefore, we developed the novel model presented in this study, which exhibits stability for several weeks. This advanced model was specifically implemented for investigating trypanosome infections through tsetse flies. Given that the epidermal morphology, metabolic activity, and presence of skin-specific proteins in the compressed models were comparable or even superior, we did not conduct a parallel study using uncompressed models. The advantages offered by our new model made it the ideal choice for our research purposes.

2. How does the mechanical compression influence the gene expression of the skin model?

Answer: Currently, we do not possess a complete dataset for the standard models. However, it can be assumed that if gene expression is affected by compression, it would likely exhibit comparable patterns in both the standard and compressed models. This assumption is based on the fact that cell-mediated contraction in the compressed models also results in a loss of water during the culture, leading to a similar collagen concentration at the end of the cultivation period.

3. Why was not a more stable alternative dermal matrix applied instead?

Answer: More stable alternatives exist. However, these models typically employ chemically modified collagens (e.g. by crosslinking) or synthetic matrices. In our previous research, we demonstrated that chemical crosslinking achieves significantly faster results compared to the reassembly of collagen fibrils. Consequently, a secondary matrix is formed that possesses stability but cannot be directly compared to native collagen (refer to Lotz and Schmid et al., ACS Appl Mater

Interfaces. 2017 21;9(24):20417-20425, "Cross-linked Collagen Hydrogel Matrix Resisting Contraction To Facilitate Full-Thickness Skin Equivalents"). Considering that an altered matrix could potentially introduce biases in the behavior of trypanosomes, we made a deliberate choice to utilize a matrix that closely resembles natural collagen. Plastic compression offered an effective means to adjust the stability of the models through a physical method without altering the chemistry of collagen. In fact, our model is mechanically exceptionally stable.

4. Modern 3D skin models try to reflect human skin in as many aspects as possible. Immune cells, such as macrophages, dendritic cells, or vascular cells, among others, are incorporated here to allow cross-talk with dermal fibroblasts and epidermal keratinocytes. Why was a model used here that only contains keratinocytes and fibroblasts?

Answer: The reviewer is correct in pointing out the availability of apparently more complex models. However, we deliberately opted for a simpler model initially to minimize variables in our experimental setup. It is worth noting that our model exhibits superior mechanical accuracy compared to other systems, which is crucial for vector infection studies. Currently, we are actively incorporating immune components and vascularization into our model, but this is a long-term project due to the demanding process of qualifying each additional level of complexity.

5. The disadvantages of an in vitro skin model compared to the in vivo situation/animal model should also be discussed.

Answer: We concur with the reviewer's assessment that skin models possess a defined scope of applicability. Specifically, for the research question tackled in this study, we firmly believe that the implemented models were better suited than in vivo skin. However, we have taken the reviewer's suggestion into account and have included the following paragraph on page 13, lines 3 – 8 to explicitly clarify that the suitability of skin models may vary for other inquiries.

" However, it is important to acknowledge that these models do not possess the same level of complexity as native human skin in terms of immune competency, vascularization, and innervation. Furthermore, the in vitro models lack the systemic component. In contrast, in vivo animal models provide these essential components, albeit with species-specific variations. Therefore, for future

studies, it is crucial to carefully consider whether an in vitro or in vivo model is more appropriate for the specific research question at hand."

REVIEWERS' COMMENTS

Reviewer #1 (Remarks to the Author):

I am pleased with the answers to my comments and those of the other reviewers. Also, more experiments were added and many modifications were made to the figures which in my opinion deserve publication in Nature Communications.

Reviewer #3 (Remarks to the Author):

In my view, the authors addressed my concerns and those of the other reviewers in a sufficient way. The additional experimental work/results that are included, is certainly of added value that improves the manuscript.

Reviewer #4 (Remarks to the Author):

No further comments/remarks

Reviewer #5 (Remarks to the Author):

The concerns of Reviewer #2 have been satisfactorily addressed in the revised manuscript.